# Optimal Neural Compressors for the Rate-Distortion-Perception Tradeoff

**Eric Lei**[†][*], **Hamed Hassani**[‡], **Shirin Saeedi Bidokhti**[‡]

[†]JPMorganChase Global Technology Applied Research, [‡]University of Pennsylvania

eric.lei@jpmchase.com, {hassani, saeedi}@seas.upenn.edu

## Abstract

Recent efforts in neural compression have focused on the rate-distortion-perception (RDP) tradeoff, where the perception constraint ensures the source and reconstruction distributions are close in terms of a statistical divergence. Theoretical work on RDP describes properties of RDP-optimal compressors without providing constructive and low complexity solutions. While classical rate-distortion theory shows that optimal compressors should efficiently pack space, RDP theory additionally shows that infinite randomness shared between the encoder and decoder may be necessary for RDP optimality. In this paper, we propose neural compressors that are low complexity and benefit from high packing efficiency through lattice coding and shared randomness through shared dithering over the lattice cells. For two important settings, namely infinite shared and zero shared randomness, we analyze the RDP tradeoff achieved by our proposed neural compressors and show optimality in both cases. Experimentally, we investigate the roles that these two components of our design, lattice coding and randomness, play in the performance of neural compressors on synthetic and real-world data. We observe that performance improves with more shared randomness and better lattice packing.

## 1   Introduction

Neural compressors learned from large-scale datasets have achieved state-of-the-art performance in terms of the rate-distortion tradeoff (Ballé et al., 2020; Yang et al., 2023), especially when trained to produce reconstructions that align well with human perception (Mentzer et al., 2020; Tschannen et al., 2018; Agustsson et al., 2019; Muckley et al., 2023). To achieve this, an additional perception loss term is used, typically defined as a statistical divergence $\delta$ between the reconstruction and source distributions. As such, recent focus has shifted to the rate-distortion-perception (RDP) framework, where compressors explore a triple tradeoff between rate, distortion and perception $\delta$ (Blau and Michaeli, 2019). The RDP function of a source $X \sim P_X$, defined as

$$
R(D, P) = \min_{P_{\hat{X}|X}} \quad I(X; \hat{X})
$$
$$
\text{s.t.} \quad \mathbb{E}_{P_X P_{\hat{X}|X}}[\Delta(X, \hat{X})] \leq D, \quad \delta(P_X, P_{\hat{X}}) \leq P, \tag{1}
$$

where $\Delta$ is a distortion function, has emerged to describe this fundamental tradeoff (Matsumoto, 2018; Blau and Michaeli, 2019; Li et al., 2011). Several RDP coding theorems have recently been proven (Theis and Wagner, 2021; Wagner, 2022; Chen et al., 2022), providing an operational meaning to (1) as a fundamental limit of lossy compression for the RDP tradeoff[1].

In this paper, we investigate how neural compressors may achieve RDP optimality, and what components are necessary for good RDP performance. The RDP coding theorems, while non-constructive,

---

[*]Prepared prior to employment at JPMorganChase.

[1]Specifically, (1) is achievable by a sequence of source codes, and no source code can do better than (1).

39th Conference on Neural Information Processing Systems (NeurIPS 2025).

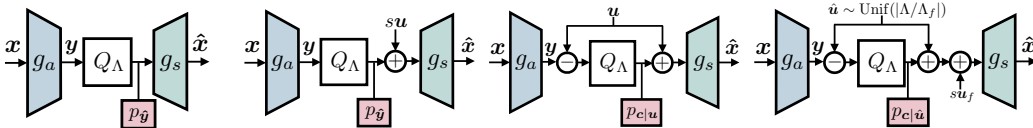

(a) Lattice transform coding (LTC); deterministic.
(b) Private-dither LTC (PD-LTC); $R_c = 0$.
(c) Shared-dither LTC (SD-LTC); $R_c = \infty$.
(d) Quantized-shared-dither LTC (QSD-LTC); finite $R_c$.

Figure 1: Lattice transform coding (LTC) with $R_c$ bits of (shared) randomness using dithering; $\boldsymbol{u} \sim \mathrm{Unif}(\mathcal{V}_0(\Lambda))$ and $\boldsymbol{u}_f \sim \mathrm{Unif}(\mathcal{V}_0(\Lambda_f))$ are continuous, and $\hat{\boldsymbol{u}} \sim \mathrm{Unif}(|\Lambda/\Lambda_f|)$ is discrete, where $\Lambda, \Lambda_f$ are nested lattices. LTC/PD-LTC entropy-code $Q_\Lambda(\boldsymbol{y})$ with likelihoods $p_{\hat{\boldsymbol{y}}}$. SD-LTC and QSD-LTC entropy-code $Q_\Lambda(\boldsymbol{y} - \boldsymbol{u})$ and $Q_\Lambda(\boldsymbol{y} - \hat{\boldsymbol{u}})$ with likelihoods $p_{\boldsymbol{c}|\boldsymbol{u}}$ and $p_{\boldsymbol{c}|\hat{\boldsymbol{u}}}$, respectively.

shed light on properties of RDP-optimal compressors. In contrast to the classical rate-distortion function $R(D) = R(D, \infty)$, which is asymptotically achievable by fully deterministic codes, achieving the RDP function may require not only stochastic encoding/decoding but also *infinite* randomness shared between the encoder and decoder. Devising neural compressors that may achieve RDP optimality at low complexity is an important step in advancing the theory and practice of neural compression. Moreover, infinite shared randomness may not always be available. In settings where shared randomness is *limited*, or *unavailable*, we are still interested in the best possible schemes.

Infinite shared randomness has previously proved successful in RDP-oriented compressors such as Theis et al. (2022) that are based on reverse channel coding (RCC) (Theis and Ahmed, 2022; Li and El Gamal, 2018; Cuff, 2013). RCC enables communication of a sample from a prescribed distribution (e.g., one that is good for RDP) under a limited rate constraint. The performance of RCC is provably near-optimal, but this comes at the cost of high complexity. Moreover, RCC heavily relies on infinite amount of randomness and does not allow for limited or zero randomness. We seek to develop methods with much lower complexity and allow for zero, limited, or infinite amount of randomness.

In the classical rate-distortion framework, recent work has investigated whether neural compressors are optimal, where vector quantization (VQ) (Gersho and Gray, 2012) is known to be optimal, but suffers from high complexity. In Lei et al. (2025), it was shown that lattice transform coding (LTC), which uses lattice quantization (LQ) in the latent space, can achieve performance close to VQ at significantly lower complexity; see also Zhang and Wu (2023); Kudo et al. (2023). VQ and LTC provide near-optimal RD performance due to high space-packing efficiency. However, it is not clear at first glance whether VQ-like coding is good for the RDP setting, where randomized reconstructions are required to satisfy the perception constraint (Tschannen et al., 2018). It is further unclear how randomness should be incorporated with quantization in a way that is RDP-optimal.

Dithering is a common method of randomizing a quantizer. Classically, a shared random dither can help quantization noise admit desired statistical properties, and has applications such as universal quantization (Ziv, 1985), and practical training methods for neural compression (Ballé et al., 2020). For the RDP setting, using dithering to introduce randomness has been investigated recently; Theis and Agustsson (2021) show on a simple source how dithered scalar quantization (SQ) benefits RDP.

In this paper, we introduce randomness, shared and private, into neural compressors via architectures that build on LTC (Fig. 1), and investigate the roles that randomness and quantization play in how neural compressors perform for the RDP tradeoff, while managing complexity. We study LTC where randomness takes the form of a random lattice dither vector. When the randomness is private (i.e., none shared), the dither is only added at the decoder (PD-LTC; Fig. 1b). When the randomness is shared, the dither is subtracted at the encoder, and added back at the decoder (SD-LTC and QSD-LTC; Figs. 1c, 1d respectively). This work unifies transform coding, lattice coding, and randomized coding, identifying the roles each play when used together for RDP. Our contributions are the following[2].

1. We propose LTC with infinite or no shared randomness, using a shared dither (SD-LTC) or private dither (PD-LTC) respectively, and describe the benefits of the former in Sec. 3. We then propose a discrete dithering scheme defined via nested lattices that allows for *finite* randomness to be shared between the encoder and decoder. This scheme, QSD-LTC, interpolates between PD- and SD-LTC, enabling control over the rate of shared randomness.

2. We theoretically analyze and show optimality of SD-LTC and PD-LTC on the Gaussian source with squared 2-Wasserstein perception and asymptotic blocklength in Sec. 4. For the former

[2]Code can be found at https://github.com/leieric/LTC-RDP.

(where infinite shared randomness is available), we use the sphere-like behavior of lattice cells and the AWGN-like behavior of dithered LQ to show that SD-LTC achieves the RDP function $R(D, P)$. For the latter (where no shared randomness is available), we use lattice Gaussian coding methods to show that under a perception constraint of $P = 0$, PD-LTC achieves $R(D/2, \infty)$, which coincides with the fundamental RDP limit under no shared randomness.

3. We empirically study PD-LTC, SD-LTC, and QSD-LTC performance on synthetic and real-world sources in Sec. 5. We verify our theory and show that RDP performance improves with increased shared randomness and better lattice packing efficiency.

## 2 Background and Related Work

**Neural Compression for RDP.** Most neural compressor designs that account for perception are derived from the nonlinear transform coding (NTC) setup (Ballé et al., 2020). These models are parameterized by analysis transform $g_a$, synthesis transform $g_s$, and entropy model $p_{\hat{y}}$; see Fig. 1a. To compress a source $x$, the encoder computes the latent $y = g_a(x)$, which gets scalar quantized via rounding. The codeword or quantized latent $\hat{y} = Q_\Lambda(y)$ is then entropy coded using an entropy model $p_{\hat{y}}$. The decoder provides the reconstruction $\hat{x} = g_s(\hat{y})$. The model is trained end-to-end via

$$\min_\theta \mathbb{E}[-\log p_{\hat{y}}(\hat{y})] + \lambda_1 \mathbb{E}[\Delta(x, \hat{x})] + \lambda_2 \delta(P_x, P_{\hat{x}}), \tag{2}$$

where $\theta$ denotes the parameters of the codec $(g_a, g_s, p_{\hat{y}})$, and $\lambda_1, \lambda_2 \geq 0$ control the RDP tradeoff. Many state-of-the-art methods (Tschannen et al., 2018; Agustsson et al., 2019; Mentzer et al., 2020; Muckley et al., 2023; He et al., 2022; Zhang et al., 2021) optimize (2), primarily differing in the choice of $\Delta$ and the way $\delta$ is estimated in practice, which is typically done using an adversarial loss involving a discriminator neural network. The use of randomness (shared or not) in these methods has not always been consistent, nor fully explored in its relation to RDP optimality. While Blau and Michaeli (2019) add uniform noise to the quantized latent, Tschannen et al. (2018); Agustsson et al. (2023) concatenate noise to the quantized latent, and Mentzer et al. (2020); Muckley et al. (2023) do not use randomness at all. Zhang et al. (2021) use (shared) dithered SQ with NTC, but do not explore why or how it is good for RDP. In contrast, we study LQ with dithering under varying levels of shared randomness, and show that one needs both the improved packing efficiency of lattices along with shared lattice dithering to achieve best performance. While there exist a few RDP-oriented neural compressors that do not fit the NTC framework (Theis et al., 2022; Yang et al., 2024; Yang and Mandt, 2024) and instead leverage diffusion models, our work focuses on NTC-style neural compressors for RDP, as they remain the most pervasive type of neural compressor in use, and do not suffer from the higher complexity of RCC or diffusion models.

**Information-Theoretic Analysis of RDP.** The RDP function in (1) was formally introduced and widely adopted following Blau and Michaeli (2019); see also earlier related work by Matsumoto (2018); Li et al. (2011); Saldi et al. (2015). (1) is a purely informational quantity, i.e., a function of the source $P_X$, and thus does not have a meaning as a fundamental limit of compression without a corresponding coding theorem describing it as such. The first RDP coding theorem was provided by Saldi et al. (2015), who show that for perfect perception ($P = 0$), (1) is achievable under infinite shared randomness, and conversely that no compressor can outperform (1). For general $P$, Theis and Wagner (2021) establish optimality (i.e., achievability and converse) of (1) when infinite shared randomness is available. Saldi et al. (2015) further characterize the fundamental RDP limit at $P = 0$ when only $R_c$ bits per sample of shared randomness is allowed, which only coincides with (1) when $R_c = \infty$; a smaller $R_c$ results in a strictly worse fundamental limit (see also Wagner (2022)). This establishes the necessity of infinite shared randomness to achieve (1). Li et al. (2011) use dithered LQ to show achievability of (1) for $P = 0$; in contrast, we show this to be true for general $P$ and also analyze the $R_c = 0$ case, which requires a new set of proof techniques based on lattice Gaussian coding (Ling and Belfiore, 2014). Under private randomness ($R_c = 0$), Yan et al. (2021) establish $R(D/2, \infty)$ as the fundamental limit for $P = 0$; Hamdi et al. (2024a) show that randomized encoders do not help. Chen et al. (2022) study the RDP tradeoff when the perception constraint is strong- or weak-sense[3]. Under weak-sense, it was shown that $R(D, P)$ is achievable without shared randomness, whereas under strong-sense, shared randomness is necessary, agreeing with Saldi et al. (2015); Wagner (2022). In our work, we focus on strong-sense, since that is typically how the

---

[3]On vectors $x, \hat{x} \in \mathbb{R}^n$, strong-sense denotes $\delta(P_x, P_{\hat{x}}) < P$; weak-sense denotes $\delta(P_{x_i}, P_{\hat{x}_i}) < P, \forall i$.

perception is measured in practice (i.e., in (2)). In particular, we focus on strong-sense Wasserstein, since that aligns with $\delta$'s and evaluation metrics chosen in practice such as Fréchet inception distance.

Coding theorems use schemes that are typically not constructive (e.g., random coding) and/or not practical (e.g., high complexity RCC schemes). They do, however, provide insights on structures that may be useful or even necessary for optimality. In addition to the necessity of infinite shared randomness to achieve (1), we show in Sec. A how the RCC scheme of Theis and Wagner (2021) implies that a good RDP compressor should behave like a randomized VQ: it should have VQ-like packing efficiency (i.e., good for distortion) combined with random codewords that follow the right distribution (i.e, good for perception). In our work, the former is handled with LQ (at low complexity), while the latter is handled with dithering; our schemes are further shown optimal on Gaussians.

The benefits of (potentially shared) randomness have also been discussed outside the context of coding theorems. Tschannen et al. (2018) show how randomized decoders are necessary to achieve perfect perceptual quality. Theis and Agustsson (2021) illustrate how quantizers benefit from shared randomness on a toy circle source. Similarly, Zhou and Tian (2024) demonstrate how staggered SQ can use limited shared randomness to improve performance on the circle. In contrast, our work presents a more general approach for infinite, limited, and no shared randomness with LQ that empirically shows its benefits and is provably optimal on Gaussians.

**Lattice Quantization.** Lattice quantization (LQ) involves a lattice $\Lambda$, which consists of a countably infinite set of codebook vectors in $n$-dimensional space (Conway and Sloane, 1999; Zamir et al., 2014). We denote $Q_\Lambda(\boldsymbol{x}) := \arg\min_{\boldsymbol{\lambda} \in \Lambda} \|\boldsymbol{\lambda} - \boldsymbol{x}\|^2$ as the LQ of a vector $\boldsymbol{x} \in \mathbb{R}^n$. The fundamental cell, or Voronoi region, of the lattice is given by $\mathcal{V}_0(\Lambda) := \{\boldsymbol{x} \in \mathbb{R}^n : Q_\Lambda(\boldsymbol{x}) = \boldsymbol{0}\}$, i.e., the set of all vectors quantized to $\boldsymbol{0}$. We denote the lattice volume as $V(\Lambda) := \int_{\mathcal{V}_0(\Lambda)} d\boldsymbol{x}$, the lattice second moment as $\sigma^2(\Lambda) := \frac{1}{n} \mathbb{E}_{\boldsymbol{u} \sim \mathrm{Unif}(\mathcal{V}_0(\Lambda))}[\|\boldsymbol{u}\|^2]$, and the normalized second moment (NSM) $G(\Lambda) = \frac{\sigma^2(\Lambda)}{(V(\Lambda))^{2/n}}$. The lattice's packing efficiency can be measured by how small its NSM is; it is known that there exists sequences of lattices $\{\Lambda^{(n)}\}_{n=1}^\infty$ that achieve the sphere lower bound, i.e., $\lim_{n \to \infty} G(\Lambda^{(n)}) = \frac{1}{2\pi e}$, where the lattice cells become sphere-like (Zamir et al., 2014, Ch. 7). The closest vector problem (CVP), which finds $Q_\Lambda(\boldsymbol{x})$, is NP-hard in general, but many lattices with low NSM (e.g., $E_8$, Barnes-Wall, Leech) have efficient CVP solvers. Recently proposed polar lattices (Liu et al., 2021), which have polynomial time CVP, were shown to be sphere-bound-achieving (Liu et al., 2024). Recently, LQ was explored in neural compression (Zhang and Wu, 2023; Kudo et al., 2023) as a low-complexity method that improves the poor packing efficiency of SQ, equivalent to the integer lattice $\mathbb{Z}_n$. Lei et al. (2025) showed that NTC transforms are insufficient to overcome the poor packing efficiency of a suboptimal lattice, leading to the lattice transform coding (LTC) framework; performance improves with increased lattice packing efficiency.

# 3   Lattice Transform Coding for RDP

We seek to design compressors that are RDP-optimal given constraints on the amount of shared randomness available and are also low complexity. As mentioned in Sec. 2, we show in Sec. A that a good RDP scheme should implement randomized VQ. Dithering is a method that can enable a quantizer to be randomized. In the context of neural compression, NTC transforms are unable to generate VQ-like regions in the source space due to the limited packing efficiency of latent space SQ (Lei et al., 2025). The LTC framework, which uses latent LQ, can provide the benefits of VQ-like regions in the source space. LQ naturally supports dithering that is uniform over the lattice cell. Thus, dithered LQ emerges as a promising scheme that is both randomized and VQ-like, while maintaining low complexity. In the following, we describe how LQ with dithering can be integrated into the LTC framework and trained end-to-end. We present three architectures that handle the cases of infinite-, no-, and finite-shared randomness via a shared-, private-, and quantized shared-dither, respectively.

## 3.1   LTC with Infinite Shared Randomness

We first define the shared-dither LTC, which assumes infinite shared randomness is available between the encoder and decoder. We denote the fundamental cell $\mathcal{V}_0(\Lambda)$ as $\mathcal{V}_0$ for ease of notation.

**Definition 3.1** (Shared-Dither Lattice Transform Code (SD-LTC); Fig. 1c)**.** A SD-LTC is a triple $(g_a, g_s, \Lambda)$, with mappings $g_a, g_s$ and lattice $\Lambda$. A random dither $\boldsymbol{u} \sim \mathrm{Unif}(\mathcal{V}_0(\Lambda))$, uniform over the

lattice cell, is shared between the encoder and decoder. The SD-LTC computes the latent $\boldsymbol{y} = g_a(\boldsymbol{x})$, entropy codes $\boldsymbol{c} = Q_\Lambda(\boldsymbol{y} - \boldsymbol{u})$ at the encoder, and the decoder outputs $\hat{\boldsymbol{x}} = g_s(\boldsymbol{c} + \boldsymbol{u})$.

Since $\boldsymbol{u}$ is available at the encoder and decoder (which utilizes the availability of infinite shared randomness), the operational rate of SD-LTC is given by

$$H(\boldsymbol{c}|\boldsymbol{u}) = \mathbb{E}_{\boldsymbol{y},\boldsymbol{u}}[-\log p_{\boldsymbol{c}|\boldsymbol{u}}(\boldsymbol{c}|\boldsymbol{u}))] = \mathbb{E}_{\boldsymbol{y},\boldsymbol{u}}\left[-\log \int_{\mathcal{V}_0 + \boldsymbol{c}} p_{\boldsymbol{y}-\boldsymbol{u}|\boldsymbol{u}}(\boldsymbol{w}|\boldsymbol{u})d\boldsymbol{w}\right] \tag{3}$$

$$= \mathbb{E}_{\boldsymbol{y},\boldsymbol{u}}[-\log \mathbb{E}_{\boldsymbol{u}'}[p_{\boldsymbol{y}}(Q_\Lambda(\boldsymbol{y} - \boldsymbol{u}) + \boldsymbol{u} + \boldsymbol{u}')]], \tag{4}$$

where $\boldsymbol{u}' \sim \mathrm{Unif}(\mathcal{V}_0)$. As an alternative, we can make use of the additive channel equivalence $(Q_\Lambda(\boldsymbol{y} - \boldsymbol{u}) + \boldsymbol{u} \overset{d}{=} \boldsymbol{y} + \boldsymbol{u}_{\mathrm{eq}}$, where $\boldsymbol{u}_{\mathrm{eq}} \sim \mathrm{Unif}(\mathcal{V}_0(\Lambda)))$ (Zamir et al., 2014, Thm. 5.2.1), yielding

$$H(\boldsymbol{c}|\boldsymbol{u}) = I(\boldsymbol{y}; \boldsymbol{y} + \boldsymbol{u}) = h(\boldsymbol{y} + \boldsymbol{u}) - h(\boldsymbol{u}) = \mathbb{E}_{\boldsymbol{y},\boldsymbol{u}}[-\log p_{\boldsymbol{y}+\boldsymbol{u}}(\boldsymbol{y} + \boldsymbol{u})] - \log V(\Lambda) \tag{5}$$

$$= \mathbb{E}_{\boldsymbol{y},\boldsymbol{u}}\left[-\log \int_{\mathcal{V}_0(\Lambda)+\boldsymbol{y}+\boldsymbol{u}} p_{\boldsymbol{y}}(\boldsymbol{w})d\boldsymbol{w}\right] = \mathbb{E}_{\boldsymbol{y},\boldsymbol{u}}[-\log \mathbb{E}_{\boldsymbol{u}'}[p_{\boldsymbol{y}}(\boldsymbol{y} + \boldsymbol{u} + \boldsymbol{u}')]], \tag{6}$$

where $h(\cdot)$ denotes differential entropy, $V(\Lambda)$ is the lattice volume, and (6) holds since $p_{\boldsymbol{y}+\boldsymbol{u}}(\boldsymbol{y} + \boldsymbol{u}) = \frac{1}{V(\Lambda)} \int_{\mathcal{V}_0(\Lambda)+\boldsymbol{y}+\boldsymbol{u}} p_{\boldsymbol{y}}(\boldsymbol{w})d\boldsymbol{w}$. The objective is trained with $\min_\theta H(\boldsymbol{c}|\boldsymbol{u}) + \lambda_1 \mathbb{E}[\Delta(\boldsymbol{x}, \hat{\boldsymbol{x}})] + \lambda_2 \delta(P_{\boldsymbol{x}}, P_{\hat{\boldsymbol{x}}})$, with learned parameters $\theta$. Either (4) or (6) can be used for $H(\boldsymbol{c}|\boldsymbol{u})$; (4) requires the straight-through estimator due to non-differentiability of the quantizer. In the following, we comment on several connections between (4), (6), and other work in the neural compression literature.

**Integrating the learned $p_{\boldsymbol{y}}$.** The inner integral in (4) and (6) can be computed exactly under SQ (equivalently, $\Lambda = \mathbb{Z}^n$) using the CDF of $p_{\boldsymbol{y}}$, following Ballé et al. (2018). For general lattices, the inner integral can be estimated using Monte-Carlo following Lei et al. (2025), by sampling vectors uniform over the lattice cell (Conway and Sloane, 1984).

**Noisy proxies and operational rates.** The equivalence of (4) and (6) was shown in Ballé et al. (2020) for $\Lambda = \mathbb{Z}_n$. Their equivalence for general lattices, as shown above, follows from Zamir et al. (2014). Ballé et al. (2020) uses this equivalence to support (6) as a training objective, which is known as the noisy proxy to quantization in the literature. This is useful since (6) is differentiable with respect to $\boldsymbol{y}$, whereas (4) is not. Here, we emphasize that both (4) and (6) represent the operational rate for a SD-LTC. For deterministic NTC/LTC trained with the noisy proxy, a deterministic dither (perhaps $\boldsymbol{0}$) is chosen test time, resulting in a different operational rate that does not average over $\boldsymbol{u}$. Agustsson and Theis (2020) proposed dithered SQ as universal quantization; however, they were motivated by reducing train/test rate mismatch rather than the RDP tradeoff, which is the focus of our work.

### 3.2 LTC with No Shared Randomness

We now consider when no shared randomness is available. As mentioned, decoder randomness is needed for the perception constraint, and this manifests itself as a private dither at the decoder.

**Definition 3.2** (Private-Dither Lattice Transform Code (PD-LTC); Fig. 1b). A PD-LTC is a triple $(g_a, g_s, \Lambda)$, with transforms $g_a$ and $g_s$ and lattice $\Lambda$. A random dither $\boldsymbol{u} \sim \mathrm{Unif}(\mathcal{V}_0(\Lambda))$, uniform over the lattice cell, is at the decoder only. The PD-LTC entropy-codes $\hat{\boldsymbol{y}} = Q_\Lambda(g_a(\boldsymbol{x}))$, and the decoder outputs $\hat{\boldsymbol{x}} = g_s(\hat{\boldsymbol{y}} + s\boldsymbol{u})$, where $s > 0$ is a parameter that controls the dither magnitude.

For a PD-LTC (shown in Fig. 1b), the operational rate is the same as in deterministic LTC, given by $H(\hat{\boldsymbol{y}}) = \mathbb{E}_{\boldsymbol{y}}[-\log p_{\hat{\boldsymbol{y}}}(\hat{\boldsymbol{y}})]$, as there is no shared dither. Therefore, the training objective remains the same as (2). The following proposition provides some intuition on why shared-dither quantization (Fig. 1c) is superior to private-dither quantization (Fig. 1b) in terms of distortion.

**Proposition 3.3.** Define $\hat{\boldsymbol{x}}_{\mathsf{SD}} = Q_\Lambda(\boldsymbol{x} - \boldsymbol{u}) + \boldsymbol{u}$, $\hat{\boldsymbol{x}}_{\mathsf{PD}} = Q_\Lambda(\boldsymbol{x}) + s\boldsymbol{u}$. For any $\boldsymbol{x}$ and $s \geq 1$,

$$\mathbb{E}\big[\|\boldsymbol{x} - \hat{\boldsymbol{x}}_{\mathsf{PD}}\|^2\big] = s^2 \mathbb{E}\big[\|\boldsymbol{x} - \hat{\boldsymbol{x}}_{\mathsf{SD}}\|^2\big] + \mathbb{E}\big[\|\boldsymbol{x} - Q_\Lambda(\boldsymbol{x})\|^2\big] \geq \mathbb{E}\big[\|\boldsymbol{x} - \hat{\boldsymbol{x}}_{\mathsf{SD}}\|^2\big]. \tag{7}$$

The proof is provided in Sec. D; note that $\boldsymbol{x}$ here is an arbitrary vector (not necessarily the source). Prop. 3.3 implies the PD error is the sum of the SD error and the error under deterministic quantization; see Fig. 2. While $\hat{\boldsymbol{x}}_{\mathsf{SD}}$ is random over the blue cell, incurring error equal to the lattice second moment, $\hat{\boldsymbol{x}}_{\mathsf{PD}}$ is random over the gray cell, incurring additional quantization error. Regarding $s \geq 1$, lower PD error is possible with $s < 1$, but the support of $\hat{\boldsymbol{y}} + s\boldsymbol{u}$ would have gaps within each lattice cell, which may make the perception constraint difficult to satisfy. In Sec. 4, we show that $s > 1$ is necessary for PD-LTC to achieve optimality on the Gaussian source.

While Prop. 3.3 provides an idea of how SD-LTC and PD-LTC may perform distortion-wise, the rate and perception are more complicated. For perception, the induced reconstruction distributions are different, since that of $\hat{x}_{\mathsf{SD}}$ is a convolution between the source and a zero-mean dither, whereas that of $\hat{x}_{\mathsf{PD}}$ is a mixture of dithers centered on lattice vectors. For the rate, compared to $H(\hat{y})$, $H(c|u)$ has additional randomness due to the averaging over $u$, and thus we may expect the rate of SD-LTC to be larger than that of PD-LTC if they share the same transforms. Ideally, this potential increase in rate can help achieve an overall superior RDP tradeoff for SD-LTC. In Sec. 4, we show that this is true on the Gaussian source, and verify it empirically on real-world sources as well in Sec. 5.

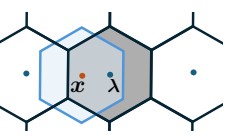

Figure 2: Reconstr. of $x$ under PD (gray) or SD (blue); $s = 1$.

### 3.3 LTC with Finite Shared Randomness

While the availability of truly infinite shared randomness may be difficult to obtain in practice, the availability of finite shared randomness may be more feasible. Let $R_c$ denote the rate of shared randomness in bits per dimension. RDP theory informs us that performance should improve as $R_c$ increases (Sec. 2). To make use of finite shared randomness, we propose a scheme (Fig. 1d) that interpolates between SD-LTC and PD-LTC by using nested lattices (Zamir et al., 2014, Ch. 8). Lattices $\Lambda, \Lambda_f$ are nested if $\Lambda$ is a sub-lattice of the fine lattice $\Lambda_f$. We restrict our attention to self-similar nested lattices $\Lambda = a\Lambda_f$ where $a > 0$ is an integer. We denote $\Lambda/\Lambda_f = \{\lambda \in \Lambda_f : \lambda \in \mathcal{V}_0(\Lambda)\}$ as the fine lattice vectors contained in the fundamental cell of $\Lambda$.

**Definition 3.4** (Quantized Shared-Dither Lattice Transform Code (QSD-LTC); Fig. 1d). A QSD-LTC is given by $(g_a, g_s, \Lambda, \Lambda_f)$, with mappings $g_a$, $g_s$ and nested lattices $\Lambda, \Lambda_f$. A random discrete dither $\hat{u} \sim \mathrm{Unif}(\Lambda/\Lambda_f)$ is shared between the encoder and decoder, and a continuous dither $u_f \sim \mathrm{Unif}(\mathcal{V}_0(\Lambda_f))$ uniform over $\Lambda_f$'s cell is at the decoder only. The QSD-LTC computes $y = g_a(x)$, entropy codes $c = Q_\Lambda(y - \hat{u})$, and the decoder outputs $\hat{x} = g_s(c + \hat{u} + su_f)$. The rate of shared randomness is $R_c = \frac{1}{n}\log|\Lambda/\Lambda_f|$ bits, i.e., there are $2^{nR_c}$ possible shared dither vectors.

**Remark 3.5.** QSD-LTC recovers PD-LTC and SD-LTC when $R_c = 0$ and $R_c = \infty$, respectively:

- When $R_c = 0$, we have that $\Lambda_f = \Lambda$, $\hat{u} = 0$, and $u_f \sim \mathrm{Unif}(\mathcal{V}_0(\Lambda))$, and therefore
$$Q_\Lambda(y - \hat{u}) + \hat{u} + su_f = Q_\Lambda(y) + su_f. \tag{8}$$
- When $R_c = \infty$, we have that $\Lambda_f = \mathbb{R}^n$, $\hat{u} \sim \mathrm{Unif}(\mathcal{V}_0(\Lambda))$, $u_f = 0$, and therefore
$$Q_\Lambda(y - \hat{u}) + \hat{u} + su_f = Q_\Lambda(y - \hat{u}) + \hat{u}. \tag{9}$$

The operational rate, $H(c|\hat{u})$, follows (4), except replacing $u$ with $\hat{u}$. The training objective is $\min_\theta H(c|\hat{u}) + \lambda_1 \mathbb{E}[\Delta(x, \hat{x})] + \lambda_2 \delta(P_x, P_{\hat{x}})$. Unlike SD-LTC, the additive channel equivalence does not apply, since the support of $y - \hat{u} - Q_\Lambda(y - \hat{u})$ is random and does not always equal $\Lambda/\Lambda_f$.

We note that the $R_c$ values that QSD-LTC may achieve are limited to $\log \Gamma$, where $\Gamma \in \mathbb{Z}^+$, a positive integer, is the nesting ratio of $\Lambda, \Lambda_f$ (Zamir et al., 2014, Ch. 8). This is due to the structure of nested lattices. To achieve $R_c$ values between 0 and 1, a non-uniform distribution for the shared dither vector $\hat{u}$ would need to be employed; we leave this to future work.

**Remark 3.6.** One may ask whether infinite shared randomness can be obtained by sending a pseudorandom seed and drawing continuous dither vectors $u \sim \mathrm{Unif}(\mathcal{V}_0(\Lambda))$ from a random number generator (RNG) based on the seed. If one compresses a source realization $x$ to a bitstream $b$, a pseudorandom seed of $k$ bits would imply that only $2^k$ possible dither vectors could be used at the decoder to decode $b$; this is noted by Hamdi et al. (2024b). Therefore, sending a pseudorandom seed is insufficient to simulate infinite shared randomness; rather, it implements a scheme with finite shared randomness. Furthermore, finite shared randomness with a constant number of bits *per dimension* is necessary to achieve the fundamental limits (Wagner, 2022); this is impossible to satisfy with a random seed of a fixed number of $k$ bits for high-dimensional sources. In addition to having the capability of imposing a $R_c$ bits per dimension of shared randomness, QSD-LTC has the additional advantage of ensuring the dither vectors are drawn uniformly from the fine lattice vectors in the lattice cell $\mathcal{V}_0(\Lambda)$. These are spread out uniformly throughout $\mathcal{V}_0(\Lambda)$ due to the structure of nested lattices. In comparison, dither vectors drawn from a random seed have no guarantee on where they may land in $\mathcal{V}_0(\Lambda)$, and would depend on the RNG and seed used. As an example, for a poorly chosen seed and RNG, the dither vectors generated could all be concentrated near the center of $\mathcal{V}_0(\Lambda)$, which would effectively yield no shared randomness. Therefore, in settings where infinite shared randomness is impractical, QSD-LTC enables a structured way of using finite shared randomness.

**On complexity.** SD-, PD-, and QSD-LTC rely on LQ. For a fixed lattice up to dimension 24 (Sec. 2), the closest codebook vector search (also used to generate dithers) can be performed at any rate in a fixed number of operations, and is significantly faster than that of VQ, which is exponential in the rate. For higher dimensions, polar lattices have complexity polynomial in the dimension.

## 4 Achieving the Fundamental Limits

We now theoretically analyze the performance of SD-LTC and PD-LTC on the Gaussian source, and describe the RDP tradeoff asymptotically achievable by SD-LTC and PD-LTC. While the operational rate and distortion are given by the per-dimension versions of those in Sec. 3, the operational perception used is per-dimension squared 2-Wasserstein distance. We leave full proofs to Sec. D.

### 4.1 Infinite Shared Randomness

**Proposition 4.1** (RDP function for Gaussian source (Zhang et al., 2021)). *Let* $P_X = \mathcal{N}(0, \sigma^2)$, $\Delta(x, \hat{x}) = (x - \hat{x})^2$, *and* $\delta(\mu, \nu) = W_2^2(\mu, \nu)$ *be squared 2-Wasserstein distance. Then*

$$R(D, P) = \begin{cases} \frac{1}{2} \log \frac{\sigma^2(\sigma - \sqrt{P})^2}{\sigma^2(\sigma - \sqrt{P})^2 - \left(\frac{\sigma^2 + (\sigma - \sqrt{P})^2 - D}{2}\right)^2}, & \text{for } \sqrt{P} < \sigma - \sqrt{|\sigma^2 - D|}, \\ \max\left\{\frac{1}{2} \log \frac{\sigma^2}{D}, 0\right\}, & \text{for } \sqrt{P} \geq \sigma - \sqrt{|\sigma^2 - D|}. \end{cases} \tag{10}$$

**Remark 4.2.** When $\sqrt{P} < \sigma - \sqrt{|\sigma^2 - D|}$, the optimal $\hat{X}$ in (1) is jointly Gaussian with marginal $\hat{X} \sim \mathcal{N}(0, (\sigma - \sqrt{P})^2)$, and covariance $\theta = \max\left\{\frac{1}{2}(\sigma^2 + (\sigma - \sqrt{P})^2 - D), 0\right\}$. When $\sqrt{P} \geq \sigma - \sqrt{|\sigma^2 - D|}$, $\hat{X}$ is jointly Gaussian with marginal $\hat{X} \sim \mathcal{N}(0, \sigma^2 - D)$.

The following theorem shows that $R(D, P)$ is achievable with SD-LTCs, and is an extension of Li et al. (2011), whose authors addressed the $P = 0$ case.

**Theorem 4.3** (Optimality of SD-LTC for Gaussian source). *Let* $X_1, X_2, \ldots \overset{i.i.d.}{\sim} \mathcal{N}(0, \sigma^2)$. *For any* $P \in [0, \sigma^2], D \in [0, 2\sigma^2]$, *there exists a sequence of SD-LTCs* $\{(g_a^{(n)}, g_s^{(n)}, \Lambda^{(n)})\}_{n=1}^{\infty}$ *such that*

$$\lim_{n \to \infty} \frac{1}{n} H(Q_{\Lambda^{(n)}}(g_a^{(n)}(X^n) - \boldsymbol{u})|\boldsymbol{u}) = R(D, P), \tag{11}$$

$$\lim_{n \to \infty} \frac{1}{n} \mathbb{E}\left[\|X^n - \hat{X}^n\|_2^2\right] \leq D, \tag{12}$$

$$\lim_{n \to \infty} \frac{1}{n} W_2^2(P_{X^n}, P_{\hat{X}^n}) \leq P, \tag{13}$$

*where* $\hat{X}^n = g_s^{(n)}(Q_{\Lambda^{(n)}}(g_a^{(n)}(X^n) - \boldsymbol{u}) + \boldsymbol{u})$, *and* $\boldsymbol{u} \sim \text{Unif}(\mathcal{V}_0(\Lambda^{(n)}))$.

**Remark 4.4.** The proof of Thm. 4.3 relies on a sphere-bound-achieving sequence of lattices with scalar transforms $g_a, g_s$. As dimension grows, the dither $\boldsymbol{u}$ becomes Gaussian-like, and the latent dithered LQ acts like an additive Gaussian channel (Zamir and Feder, 1996), imposing a joint Gaussian relationship between $X$ and $\hat{X}$ as desired by the RDP solution; see Remark 4.2 and Fig. 3.

### 4.2 No Shared Randomness

We now consider the case when the encoder and decoder do not have access to any shared randomness. For simplicity and ease of presentation, we consider the regime of near-perfect and perfect perception, corresponding to $P < \epsilon$ for any $\epsilon > 0$ and $P = 0$, respectively. This is the only regime of perception where lower bounds on the RDP achievable under no shared randomness are known (Saldi et al., 2015; Wagner, 2022; Chen et al., 2022; Yan et al., 2021), which corresponds to $R(D/2, \infty)$, and evaluates to $\frac{1}{2} \log \frac{2\sigma^2}{D}$ on the i.i.d. Gaussian source. The following theorem shows that $R(D/2, \infty)$ is achievable with PD-LTCs under near-perfect perception, which can be easily extended to perfect perception (see Remark D.8). As discussed in Sec. 2, $R(D, 0) < R(D/2, \infty)$; see Fig. 4 for a visualization.

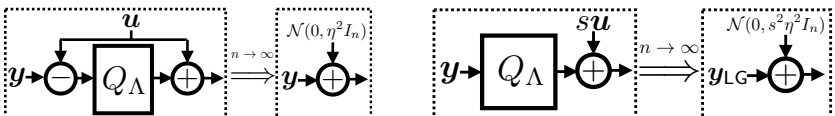

Figure 3: In the latent space, SD-LTC (left) becomes AWGN-like. PD-LTC (right) models the sum of a lattice Gaussian and Gaussian; $s$ must be large enough for the sum to be Gaussian.

**Theorem 4.5** (Optimality of PD-LTC for Gaussian source). *Let $X_1, X_2, \ldots \overset{\text{i.i.d.}}{\sim} \mathcal{N}(0, \sigma^2)$. For any $D$ satisfying $0 < D \leq 2\sigma^2$, there exists a sequence of PD-LTCs $\{(g_a^{(n)}, g_s^{(n)}, \Lambda^{(n)})\}_{n=1}^{\infty}$ such that*

$$\lim_{n \to \infty} \frac{1}{n} H(Q_{\Lambda^{(n)}}(g_a^{(n)}(X^n))) \leq R(D/2, \infty), \tag{14}$$

$$\lim_{n \to \infty} \frac{1}{n} \mathbb{E}[\|X^n - \hat{X}^n\|^2] \leq D, \tag{15}$$

$$\lim_{n \to \infty} \frac{1}{n} W_2^2(P_{X^n}, P_{\hat{X}^n}) = 0, \tag{16}$$

*where $\hat{X}^n = g_s^{(n)}(Q_{\Lambda^{(n)}}(g_a^{(n)}(X^n)) + s\boldsymbol{u})$, $\boldsymbol{u} \sim \text{Unif}(\mathcal{V}(\Lambda^{(n)}))$, and $s = \frac{\sigma}{\sqrt{\sigma^2 - D/2}}$.*

**Remark 4.6.** PD-LTC does not use a shared dither; we cannot make use of classical results (Zamir and Feder, 1996) to analyze the statistical behavior of latent LQ as an additive channel (as opposed to Thm. 4.3). Instead, Thm. 4.5 relies on lattice Gaussian coding methods (Ling and Belfiore, 2014); see Sec. D.3 for details. Using scalar transforms, $Q_{\Lambda}(g_a(\boldsymbol{x}))$ behaves like a lattice Gaussian. The additive private dither $s\boldsymbol{u}$ (which becomes Gaussian-like) makes the reconstruction Gaussian-like (Fig. 3). It then suffices for $g_s$ to scale $\hat{X}^n$ to impose the desired variance $\sigma^2$. In classical rate-distortion, dithered LQ and lattice Gaussian coding can both achieve optimality, so their differences are primarily practical (e.g., restrictions on lattice families, availability of a shared dither). Thms. 4.3, 4.5 show that with a perception constraint, the two proof techniques lead to different fundamental limits as well.

**Remark 4.7.** We note that the choice of $s = \sigma/\sqrt{\sigma^2 - D/2} > 1$ depends on the regime of the rate-distortion tradeoff; it is approximately 1 for small $D$, and becomes large for $D \to 2\sigma^2$. At low rates, this means the effective dither $s\boldsymbol{u}$ "leaks" outside the lattice cell rather significantly. This is required to ensure the perception term vanishes in (16). Specifically, $s\boldsymbol{u}$ needs to be large enough relative to the quantization steps of $Q(g_a^{(n)}(X^n))$ for $Q_{\Lambda^{(n)}}(g_a^{(n)}(X^n)) + s\boldsymbol{u}$ to approximate a Gaussian of covariance $\sigma^2 I_n$ arbitrarily accurately. The nature of this approximation is based on the flatness factor (Ling et al., 2014, Def. 5) used in lattice Gaussian coding, which is elaborated on in Remark D.7.

## 5 Experimental Results

**Experimental setup.** We denote $n$ the source dimension and $n_L$ the latent space dimension. We denote LTC as NTC when the lattice is chosen to be the integer lattice, i.e., $\Lambda = \mathbb{Z}_{n_L}$. We train the PD-, SD- and QSD-LTC models using their RDP objectives discussed in Sec. 3. For the rates reported at test time, we use the (4) version of $H(\boldsymbol{c}|\boldsymbol{u})$ for SD-LTC, $\mathbb{E}_{\boldsymbol{y}}[-\log p_{\hat{\boldsymbol{y}}}(\hat{\boldsymbol{y}})]$ for PD-LTC, and $H(\boldsymbol{c}|\hat{\boldsymbol{u}})$ for QSD-LTC; all require hard quantization. These rates are cross-entropy upper bounds on the true entropy, due to the learned $p_{\boldsymbol{y}}$ density. We use MSE distortion $\Delta(\boldsymbol{x}, \hat{\boldsymbol{x}}) = \frac{1}{n}\|\boldsymbol{x} - \hat{\boldsymbol{x}}\|_2^2$. For perception, to obtain reliable estimates in higher dimensions, we use squared sliced Wasserstein distance

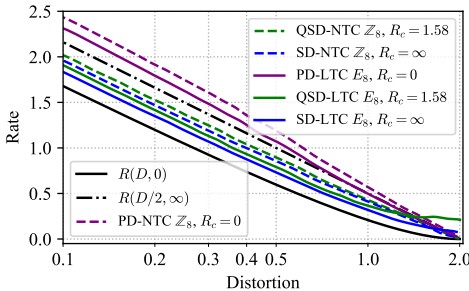

Figure 4: Effect of lattice choice and shared randomness on Gaussians at $P = 0$.

(Bonneel et al., 2015) of order 2, $\delta(P_{\boldsymbol{x}}, P_{\hat{\boldsymbol{x}}}) = \frac{1}{n}\text{SW}_2^2(P_{\boldsymbol{x}}, P_{\hat{\boldsymbol{x}}})$. During training, we use the straight-through estimator with hard quantization. See Sec. B for further details on architecture/training.

### 5.1 Synthetic Sources

We first evaluate the i.i.d. Gaussian source of dimension $n = 8$. In Fig. 4, we plot the equi-perception curves with $P = 0$ to compare the methods under the perfect realism setting. As shown, for a fixed

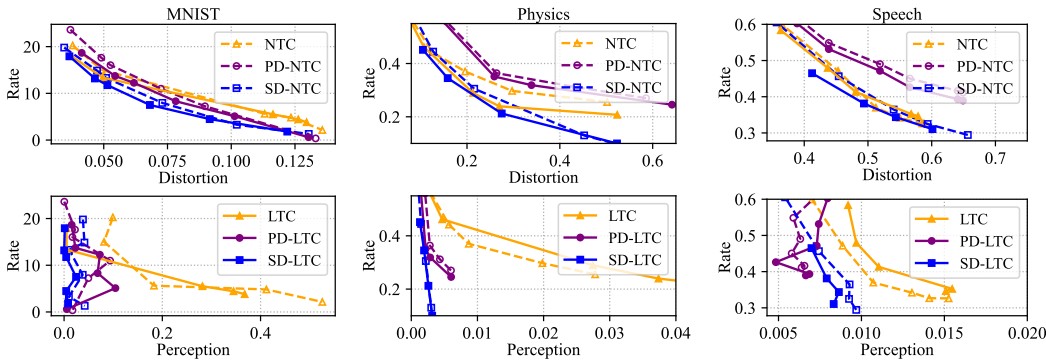

Figure 5: LTC, PD-LTC, and SD-LTC on real-world sources; R-D (top) and R-P (bottom).

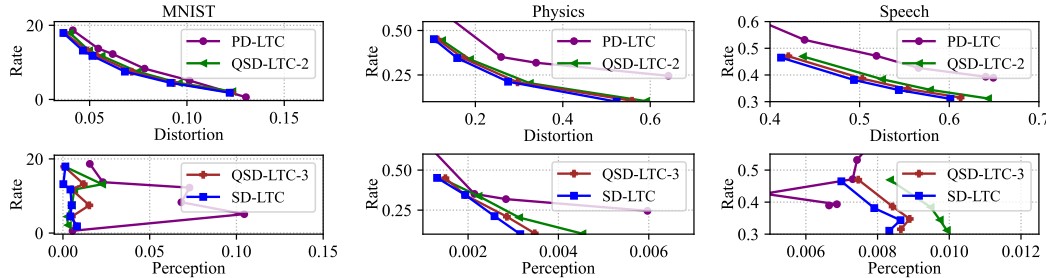

Figure 6: QSD-LTC-$\Gamma$ ($\Gamma$ the nesting ratio) on real-world sources; R-D (top) and R-P (bottom).

amount of shared randomness, a more efficient lattice improves performance. Analogously, for a fixed lattice, more shared randomness improves performance. We additionally evaluate QSD-LTC with finite shared randomness. The fine lattice $\Lambda_f$ in Def. 3.4 is set to be self-similar with $\Lambda$, with a nesting ratio of 3. This results in $R_c = \log 3$ bits per dimension. As shown, the QSD-LTC performance is nearly able to achieve that of SD-LTC, despite not having infinite shared randomness; this verifies that QSD-LTC allows one to interpolate between SD-LTC and PD-LTC. At low rates, the LTC with $E_8$ has some suboptimality; this is due to the Monte-Carlo estimation when computing the integral for latent likelihoods (Lei et al., 2025).

When compared to the fundamental limits described in Sec. 4, we see that the PD models are lower bounded by $R(D/2, \infty)$, and the SD models are are lower bounded by the RDP function $R(D, 0)$ but outperform $R(D/2, \infty)$ when the rate is not too small. Although the fundamental RDP limits should be interpreted operationally with perception measured as Wasserstein and not sliced Wasserstein, we empirically verify that on Gaussians, sliced Wasserstein is faithful to Wasserstein in Appendix. E. The performance of $n = 1$ RCC (i.e., each dimension of $\boldsymbol{x}$ is compressed with RCC separately) and $n = 8$ RCC is shown in Fig. 7a. Performance improves with increasing dimension. However, the 8-dimensional RCC is outperformed by SD-LTC, and additionally has complexity exponential in dimension and rate; SD-LTC does not suffer the same high complexity. We show the RDP achieved by deterministic NTC in Fig. 7b; at low rates the lack of randomness prevents it from enforcing the perception constraint. At larger rates, its performance coincides with SD-NTC.

## 5.2 Real-World Sources

We use MNIST (Lecun et al., 1998), Physics and Speech datasets (Yang and Mandt, 2022). These contain grayscale images of dimension $28 \times 28$, physics measurements of dimension 16, and audio signals of dimension 33 respectively. We use a latent dimension of $n_L = 8$ for the first two and $n_L = 16$ for Speech. We use the integer lattice for NTC models; for MNIST/Physics, we use the $E_8$ lattice, and for Speech, we use the $\Lambda_{16}$ lattice (Barnes and Wall, 1959). The corresponding RDP tradeoff is shown in Fig. 5. Due to lack of randomness, deterministic NTC and LTC are unable to enforce the perception constraint at lower rates, no matter how large $\lambda_2$ in (2) is set. Similar to the Gaussian case, performance improves with better lattices and increased shared randomness, demonstrating the benefits of lattices and shared randomness described in the prior sections translate to real-world sources that require nonlinear transforms. For QSD-LTC, we use self-similar nested lattices with a nesting ratios of 2 and 3, corresponding to $R_c = \log 2 = 1$ and $R_c = \log 3 \approx 1.58$

respectively. Performance increases with $R_c$ (Fig. 6). A full comparison across all models/datasets is shown in Figs. 8, 9 and 10. Overall, SD-LTC achieves the RDP tradeoffs. For a fixed lattice, QSD-LTC, with $R_c = \log 3$, can nearly achieve the performance of SD-LTC, which uses $R_c = \infty$.

### 5.3 Ablation Study

We perform an ablation study on lattice choice and training of PD-LTC with STE or the noisy proxy. We use the $D_n^*$ lattice for SD-LTC in Fig. 11 on Speech. Its performance lies between integer and Barnes-Wall lattices, which aligns with the fact that the $D_n$ packing efficiency lies between those two lattices. This supports our result in Thm. 4.3 that performance is optimal when the lattices pack space more efficiently. For the noisy proxy, we use it to train PD-LTC, but this may result in a train/test mismatch, since unlike SD-LTC, the noisy proxy does not equal the rate under hard quantization. A comparison of the two in Fig. 12 shows there is not much difference in the resulting performance.

## 6 Conclusion and Limitations

We investigate low-complexity, high-performing compressors for the RDP tradeoff. We propose combining dithered LQ with neural compression, which supports different amounts of shared randomness. Under infinite and no shared randomness, we show SD-LTC and PD-LTC achieve optimality on the Gaussian source. We empirically verify that performance improves with increased shared randomness and improved lattice efficiency across synthetic and real-world data. Future work may address: (i) expanding the range of $R_c$ in QSD-LTC, (ii) theoretical analysis of QSD-LTC, which may require new tools developed for dither vectors defined over nested lattices, and (iii) extension to SOTA image compression architectures, which would require careful integration of random dithering with hyperprior models that implicitly use a deterministic dither.

**Acknowledgements.** This work was supported by The Institute for Learning-enabled Optimization at Scale (TILOS), under award number NSF-CCF-2112665.

**Disclaimer.** This paper was prepared by Eric Lei prior to his employment at JPMorgan Chase & Co.. Therefore, this paper is not a product of the Research Department of JPMorgan Chase & Co. or its affiliates. Neither JPMorgan Chase & Co. nor any of its affiliates makes any explicit or implied representation or warranty and none of them accept any liability in connection with this paper, including, without limitation, with respect to the completeness, accuracy, or reliability of the information contained herein and the potential legal, compliance, tax, or accounting effects thereof. This document is not intended as investment research or investment advice, or as a recommendation, offer, or solicitation for the purchase or sale of any security, financial instrument, financial product or service, or to be used in any way for evaluating the merits of participating in any transaction.

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

# A   Reverse Channel Coding

## A.1   RCC Preliminaries

Reverse channel coding (RCC), also known as channel simulation, devises schemes for sending a sample from a prescribed distribution with a rate constraint (Cuff, 2013; Li, 2024), and typically assumes shared randomness. Since RDP requires randomized reconstructions, RCC can be applied to RDP, and in fact is a technique used to prove RDP coding theorems; Theis and Wagner (2021) use the one-shot RCC scheme of Li and El Gamal (2018), and Saldi et al. (2015); Wagner (2022) use channel synthesis results of Cuff (2013). We focus our discussion here on the particular RCC scheme of Li and El Gamal (2018), defined below.

**Definition A.1** (One-shot reverse channel coding (RCC) via the Poisson functional representation (Li and El Gamal, 2018)). Let $X \sim P_X$ be the source, and $P_{\hat{X}|X}$ be a channel we wish to simulate. Define $Q_{\hat{X}}$ to be the $\hat{X}$-marginal of the joint distribution $P_X P_{\hat{X}|X}$. Suppose that the same sequence of $\tau_1, \tau_2, \cdots \sim \mathrm{Exp}(1)$ and $\hat{X}_1, \hat{X}_1, \ldots \overset{\text{i.i.d.}}{\sim} Q_{\hat{X}}$ are generated at both the encoder and decoder (requiring infinite shared randomness $U$). Let $W_i = \sum_{j=1}^{i} \tau_j$. Given a source realization $X = x$, the encoder computes

$$K = \arg\min_i W_i \frac{dQ_{\hat{X}}}{dP_{\hat{X}|X}(\cdot|x)}(\hat{X}_i), \tag{17}$$

and entropy-codes it. The decoder simply outputs $\hat{X}_K$. By Li and El Gamal (2018), it holds that $\hat{X}_K|\{X = x\} \sim P_{\hat{X}|X}(\cdot|x)$. Additionally, the rate satisfies

$$H(K|U) \le I(X; \hat{X}) + \log(I(X; \hat{X}) + 1) + 4. \tag{18}$$

**Remark A.2.** The one-shot RCC technique above enables immediate achievability results for informational quantities, such as the rate-distortion-perception function $R(D, P)$, by simulating the channel $P_{\hat{X}|X}$ that achieves the infimum in (1). Due to the fact that the reconstruction $\hat{X}_K$ has distribution equal to the channel $P_{\hat{X}|X}$, any constraint in the informational quantity, such as the expected distortion constraint, or the perception constraint, is automatically satisfied by RCC. The $I(X; \hat{X})$ term in (18) then becomes equal to the informational quantity $R(D, P)$. Applying this scheme to i.i.d. blocks yields the asymptotic result. This is the approach used to prove achievability in Theis and Wagner (2021).

## A.2   RCC as Randomized VQ

The scheme in Def. A.1 essentially chooses a random codebook at both the encoder and decoder. The encoder chooses a codeword $\hat{X}_K$ according to the criterion in (17), which may appear abstract at first glance. However, the following two propositions show that when the channel $P_{\hat{X}|X}$ used to simulate is chosen to be RD- or RDP-achieving, it becomes clear that (17) uses a minimum-distance codebook search similar to VQ.

**Proposition A.3** (RCC on the rate-distortion-achieving channel; Prop. 1 of Lei et al. (2022)). *Let $P_{\hat{X}|X}$ be the rate-distortion-achieving channel, i.e., the channel achieving the infimum of $R(D, \infty)$. Then the density ratio satisfies*

$$\frac{dP_{\hat{X}|X}(\cdot|x)}{dQ_{\hat{X}}}(\hat{x}) = \frac{e^{-\beta \Delta(x, \hat{x})}}{\mathbb{E}_{\hat{X}' \sim Q_{\hat{X}}}[e^{-\beta \Delta(x, \hat{X}')}]}, \tag{19}$$

*and (17) is equivalent to*

$$K = \arg\min_i \Delta(x, \hat{X}_i) + \frac{1}{\beta} \ln W_i, \tag{20}$$

*where $\beta > 0$ is the unique Lagrange multiplier determining $I(X; \hat{X}) = R(D, \infty)$ at $D = \mathbb{E}_{P_X P_{\hat{X}|X}}[\Delta(X, \hat{X})]$.*

**Proposition A.4** (RCC on the RDP-achieving channel with $f$-divergence perception). *Assume that the perception is measured by $\delta(P_X, P_{\hat{X}}) = D_f(P_X || P_{\hat{X}})$, a $f$-divergence. Let $P_{\hat{X}|X}$ be the RDP-achieving channel that achieves* (1). *Serra et al. (2023) show that the density ratio satisfies*

$$\frac{dP_{\hat{X}|X}(\cdot|x)}{dQ_{\hat{X}}}(\hat{x}) = \frac{e^{-\beta_1 \Delta(x,\hat{x}) - \beta_2 g(P_X, Q_{\hat{X}}, \hat{x})}}{\mathbb{E}_{\hat{X}' \sim Q_{\hat{X}}}\left[e^{-\beta_1 \Delta(x,\hat{X}') - \beta_2 g(P_X, Q_{\hat{X}}, \hat{X}')}\right]}, \tag{21}$$

*where $g(P_X, Q_{\hat{X}}, \hat{x}') := f\left(\frac{dP_X}{dQ_{\hat{X}}}(\hat{x})\right) - \frac{dP_X}{dQ_{\hat{X}}}(\hat{x}) \partial f\left(\frac{dP_X}{dQ_{\hat{X}}}(\hat{x})\right)$, and $\beta_1, \beta_2 > 0$ are the unique Lagrange multipliers determining $I(X; \hat{X}) = R(D, P)$, at $D = \mathbb{E}_{P_X P_{\hat{X}|X}}[\Delta(X, \hat{X})]$ and $P = D_f(P_X || P_{\hat{X}})$. Therefore,* (17) *is equivalent to*

$$K = \arg\min_i \Delta(x, \hat{X}_i) + \frac{\beta_2}{\beta_1} g(P_X, Q_{\hat{X}}, \hat{X}_i) + \frac{1}{\beta_1} \ln W_i. \tag{22}$$

*Proof.* The proof follows Lei et al. (2022, Prop. 1), using (21) instead of (19). □

**Remark A.5.** The above two propositions imply that RCC with the RD- or RDP-achieving channels result the encoder searching for the random codeword in $\{\hat{X}_1, \hat{X}_2, \dots\}$ that is closest to the source realization $x$ in terms of the distortion metric $\Delta$, regularized by $\ln W_i$, which enforces the rate constraint; for the RDP case, it is additionally regularized by $g(P_X, Q_{\hat{X}}, \hat{X}'_i)$ which enforces the perception constraint.

Thus, when taking distortion into account, RCC can be seen as a sort of randomized VQ that finds the minimum-distance codeword (which are random). Despite its optimality for RDP, RCC is of high complexity (exponential in rate and dimension), and requires infinite shared randomness.

**Remark A.6.** A closed-form for the density ratio of the RDP-achieving channel $P_{\hat{X}|X}$ for when the perception is measured by squared 2-Wasserstein (which is the focus of our paper) is currently not known. However, we conjecture that similar to (19) and (21), it will consist of a $e^{-\beta' \Delta(x,\hat{x})}$ term in the numerator, which will result in (17) having $\Delta(x, \hat{X}_i)$ in the objective. Another slight discrepancy to our setup is that the $n$-letter operational perception in Serra et al. (2023) for the $f$-divergence perception RDP function is measured in the weak-sense (see Sec. 2). We are focused on the strong-sense setting as it more faithfully describes practical usage.

### A.3 RCC Implementation Details

To simulate RCC, one can implement the scheme in Def. A.1 and use (20) or (22) for finding the index to entropy-code. For the Gaussian source, the RDP-achieving channel and output marginal is given in Remark 4.2; we can directly implement (17) in closed-form since these distributions are Gaussian. This is what is done for the results in Fig. 7a. Since it is not possible to generate an infinite number of samples $\hat{X}_i$, we generate a codebook $N = 10,000$ samples instead. Following Li and El Gamal (2018), the index $K$ is entropy-coded using a Zipf distribution with parameter $\lambda = 1 + 1/(I(X; \hat{X} + e^{-1} \log e + 1)$. A full algorithm describing the encoding and decoding process can be found in Theis and Ahmed (2022) as well as Lei et al. (2022).

## B  Additional Experimental Details

For the synthetic (i.i.d. Gaussian source), we set $n_L = 8$, $g_a$ and $g_s$ to be linear functions, as the constructions in Thm. 4.3 and Thm. 4.5 suggest this to be sufficient for optimality, and use the $\mathbb{Z}_8$ and $E_8$ lattices. To cover the RDP tradeoff, we sweep a variety of $\lambda_1, \lambda_2$ values. For the Speech and Physics datasets, we use MLPs for $g_a$ and $g_s$ of depth 3, hidden dimension 100, and softplus nonlinearities. For MNIST, we follow the same exact experimental setup of Blau and Michaeli (2019), including model architecture, and using the test Wasserstein distance via the discriminator neural network. For NTC models, we use the factorized $p_y$ of Ballé et al. (2018), and for LTC models, we use the RealNVP normalizing flow (Dinh et al., 2017). All models are trained for 100 epochs on the

training data split, and reported metrics are averaged over the test split. Training is performed on a NVIDIA RTX5000 GPU.

For the real-world sources, we swept $\lambda_1$ (the distortion weight) and kept $\lambda_2$ (the perception weight) fixed to a positive value. This allows us to compare rate-distortion and rate-perception tradeoffs across methods; note that the two plots should be examined jointly together, and that the individual plots do not show the rate-perception (or rate-distortion) performance for a fixed distortion (or perception).

## C  Additional Empirical Results

Figures pertaining to the experimental evaluation in Sec. 5, such as ablation studies, are shown here.

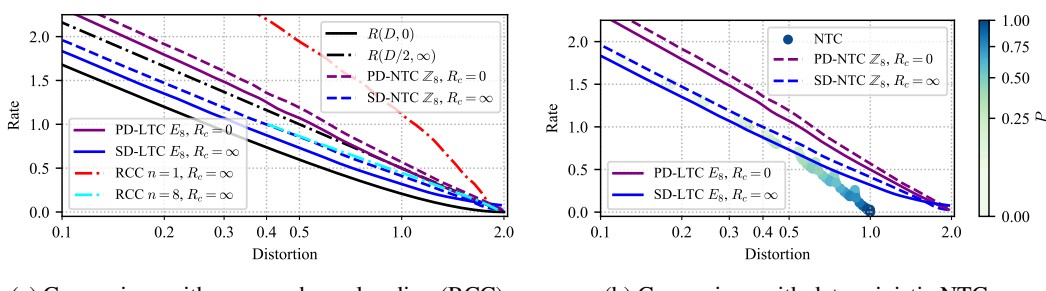

(a) Comparison with reverse channel coding (RCC).          (b) Comparison with deterministic NTC.

Figure 7: Gaussian RDP results, comparing PD-LTC and SD-LTC with RCC and deterministic NTC.

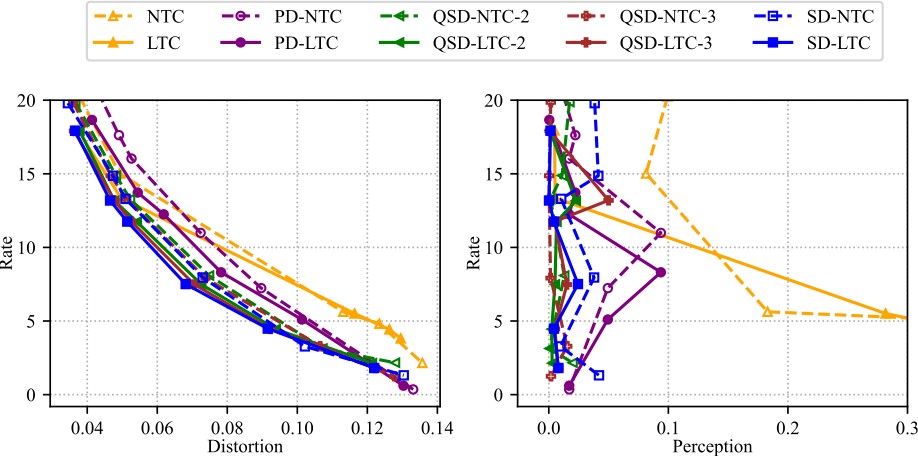

Figure 8: RDP tradeoff of all models on MNIST. QSD-NTC/LTC-$\Gamma$ corresponds to QSD-NTC/LTC with a nesting ratio of $\Gamma$.

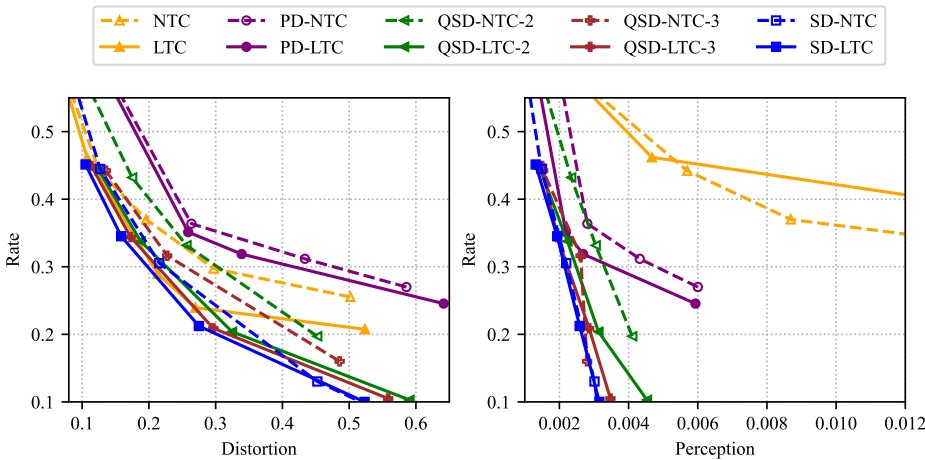

Figure 9: RDP tradeoff of all models on Physics. QSD-NTC/LTC-Γ corresponds to QSD-NTC/LTC with a nesting ratio of Γ.

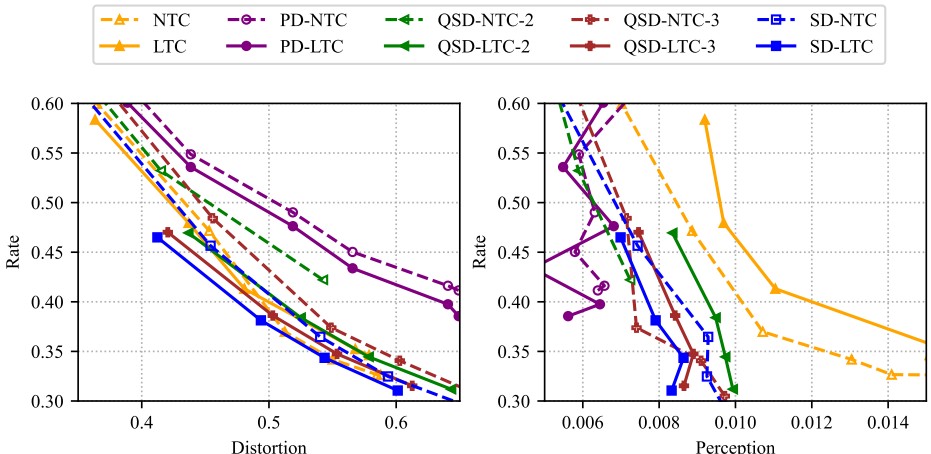

Figure 10: RDP tradeoff of all models on Speech. QSD-NTC/LTC-Γ corresponds to QSD-NTC/LTC with a nesting ratio of Γ.

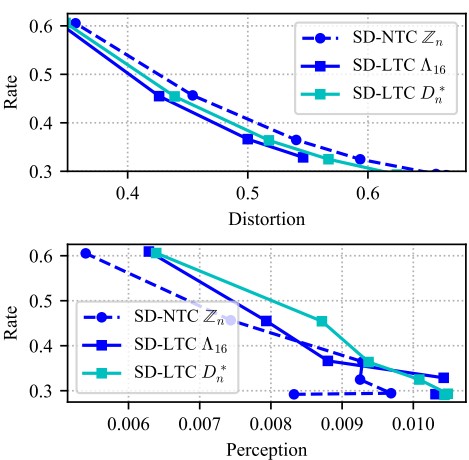

Figure 11: Comparing different lattice choices for SD-LTC, on Speech.

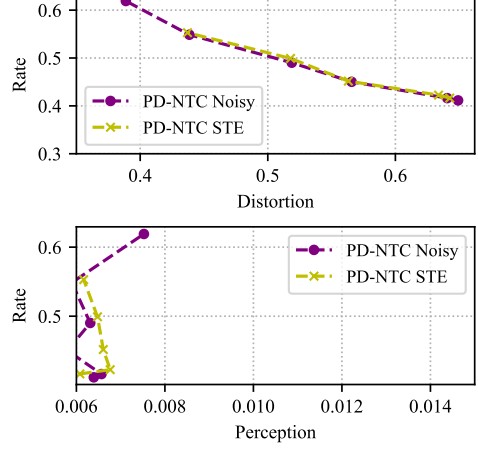

Figure 12: Comparing straight-through estimator (STE) with hard quantization vs. noisy proxy for PD-NTC, on Speech.

# D Proofs

## D.1 Proof of Proposition 3.3

*Proof.* By the crypto lemma (Zamir et al., 2014), we have that $\hat{\boldsymbol{x}}_{\text{SD}} = Q_\Lambda(\boldsymbol{x} - \boldsymbol{u}) + \boldsymbol{u} \overset{d}{=} \boldsymbol{x} + \boldsymbol{u}$. Then

$$\frac{1}{n} \mathbb{E}[\|\boldsymbol{x} - \hat{\boldsymbol{x}}_{\text{SD}}\|^2] = \frac{1}{n} \mathbb{E}[\|\boldsymbol{u}\|^2] = \sigma^2(\Lambda), \tag{23}$$

the second moment of the lattice. Additionally,

$$\frac{1}{n} \mathbb{E}[\|\boldsymbol{x} - \hat{\boldsymbol{x}}_{\text{PD}}\|^2] = \frac{1}{n} \mathbb{E}[\|\boldsymbol{x} - Q_\Lambda(\boldsymbol{x}) - s\boldsymbol{u}\|^2] \tag{24}$$

$$= \frac{1}{n} \mathbb{E}[\|\boldsymbol{x} - Q_\Lambda(\boldsymbol{x})\|^2] + \frac{1}{n} s^2 \mathbb{E}[\|\boldsymbol{u}\|^2] \tag{25}$$

$$= s^2 \sigma^2(\Lambda) + \frac{1}{n} \mathbb{E}[\|\boldsymbol{x} - Q_\Lambda(\boldsymbol{x})\|^2] \tag{26}$$

$$\geq s^2 \sigma^2(\Lambda) \tag{27}$$

$$\geq \sigma^2(\Lambda), \tag{28}$$

as desired. $\square$

## D.2 Proof of Theorem 4.3

**Theorem D.1** (Optimality of SD-LTC for Gaussian sources (Thm. 4.3 in main text)). *Let* $X_1, X_2, \ldots \overset{\text{i.i.d.}}{\sim} \mathcal{N}(0, \sigma^2)$. *For any $P$ and $D$ satisfying $0 \leq P \leq \sigma^2$ and $0 < D \leq 2\sigma^2$, there exists a sequence of SD-LTCs $\{(g_a^{(n)}, g_s^{(n)}, \Lambda^{(n)})\}_{n=1}^\infty$ such that the achieved rate, distortion, and perception satisfy*

$$\lim_{n\to\infty} \frac{1}{n} H(Q_{\Lambda^{(n)}}(g_a^{(n)}(X^n) - \boldsymbol{u})|\boldsymbol{u}) = R(D, P), \tag{29}$$

$$\lim_{n\to\infty} \frac{1}{n} \mathbb{E}[\|X^n - \hat{X}^n\|_2^2] \leq D, \tag{30}$$

$$\lim_{n\to\infty} \frac{1}{n} W_2^2(X^n, \hat{X}^n) \leq P, \tag{31}$$

*where $\hat{X}^n = g_s^{(n)}(Q_{\Lambda^{(n)}}(g_a^{(n)}(X^n) - \boldsymbol{u}) + \boldsymbol{u})$, and $\boldsymbol{u} \sim \text{Unif}(\mathcal{V}_0(\Lambda^{(n)}))$.*

*Proof.* We first focus on the case when $\sqrt{P} < \sigma - \sqrt{|\sigma^2 - D|}$. Define the sequence of DLTCs $\{(g_a^{(n)}, g_s^{(n)}, \Lambda^{(n)})\}_{n=1}^\infty$ as follows. Choose $\Lambda^{(n)}$ to be a sequence of sphere-bound-achieving lattices, i.e., $\lim_{n\to\infty} G(\Lambda^{(n)}) = \frac{1}{2\pi e}$, where $G(\cdot)$ is the normalized second moment, such that the second moment $\sigma^2(\Lambda^{(n)}) = \eta^2 := \sigma^2\left[\frac{\sigma^2(\sigma - \sqrt{P})^2}{\frac{1}{4}(\sigma^2 + (\sigma - \sqrt{P})^2 - D)^2} - 1\right]$. Set $g_a^{(n)}(\boldsymbol{v}) = \boldsymbol{v}$ to be identity mapping, and set $g_s^{(n)}(\boldsymbol{v}) = \frac{\sigma - \sqrt{P}}{\sqrt{\sigma^2 + \eta^2}}\boldsymbol{v}$.

We first verify the perception constraint is satisfied. Fix $\epsilon > 0$. From Zamir et al. (2014, Thm. 7.3.3), we have that

$$\frac{1}{n} D_{\text{KL}}(\boldsymbol{u}^{(n)}\|\boldsymbol{z}) < \frac{\epsilon^2}{2\sigma^2} \tag{32}$$

for $n$ sufficiently large, where $\boldsymbol{u}^{(n)} \sim \text{Unif}(\mathcal{V}_0(\Lambda^{(n)}))$ is uniform over the fundamental cell of $\Lambda^{(n)}$, and $\boldsymbol{z} \sim \mathcal{N}(0, \eta^2 I_n)$. Let $P_{\tilde{Y}^n} = \mathcal{N}\left(0, (\sigma - \sqrt{P})^2 I_n\right)$. Then, for $n$ sufficiently large,

$$\frac{1}{n} D_{\mathsf{KL}}(P_{\hat{X}^n} || P_{\tilde{Y}^n}) = \frac{1}{n} D_{\mathsf{KL}}\left( \frac{\sigma - \sqrt{P}}{\sqrt{\sigma^2 + \eta^2}}(X^n + \boldsymbol{u}^{(n)}) || \frac{\sigma - \sqrt{P}}{\sqrt{\sigma^2 + \eta^2}}(X^n + \boldsymbol{z}) \right) \tag{33}$$

$$= \frac{1}{n} D_{\mathsf{KL}}(X^n + \boldsymbol{u}^{(n)} || X^n + \boldsymbol{z}) \tag{34}$$

$$\leq \frac{1}{n} D_{\mathsf{KL}}(\boldsymbol{u}^{(n)} || \boldsymbol{z}) \tag{35}$$

$$\leq \frac{\epsilon^2}{2\sigma^2}, \tag{36}$$

where (33) holds by the crypto lemma (Zamir et al., 2014, Ch. 4.1), (34) holds since KL-divergence is invariant to affine transformations, and (35) is by data-processing inequality. Thus, for $n$ sufficiently large,

$$\frac{1}{\sqrt{n}} W_2\left(P_{X^n}, P_{\hat{X}^n}\right) \leq \frac{1}{\sqrt{n}} W_2(P_{X^n}, P_{\tilde{Y}^n}) + \frac{1}{\sqrt{n}} W_2\left(P_{\hat{X}^n}, P_{\tilde{Y}^n}\right) \tag{37}$$

$$\leq \frac{1}{\sqrt{n}} W_2(P_{X^n}, P_{\tilde{Y}^n}) + \frac{1}{\sqrt{n}} \cdot \sqrt{2\sigma^2 \cdot D_{\mathsf{KL}}\left(P_{\hat{X}^n} || P_{\tilde{Y}^n}\right)} \tag{38}$$

$$\leq \frac{1}{\sqrt{n}} W_2(P_{X^n}, P_{\tilde{Y}^n}) + \epsilon \tag{39}$$

$$\leq \frac{1}{\sqrt{n}} \sqrt{\sum_{i=1}^{n} W_2^2\left(P_{X_i}, \mathcal{N}\left(0, (\sigma - \sqrt{P})^2\right)\right)} + \epsilon \tag{40}$$

$$= W_2\left(\mathcal{N}(0, \sigma^2), \mathcal{N}\left(0, (\sigma - \sqrt{P})^2\right)\right) + \epsilon \tag{41}$$

$$= \sqrt{P} + \epsilon, \tag{42}$$

where (37) holds since Wasserstein distance satisfies triangle inequality, (38) is by Talagrand (1996), and (40) holds by properties of 2-Wasserstein distance on product measures (Panaretos and Zemel, 2019). By continuity of $z \mapsto z^2$, we have $\lim_{n \to \infty} \frac{1}{n} W_2^2(X^n, \hat{X}^n) \leq P$.

The rate achieved will satisfy

$$\lim_{n \to \infty} \frac{1}{n} H(Q_{\Lambda^{(n)}}(X^n - \boldsymbol{u}^{(n)}) | \boldsymbol{u}^{(n)}) = \lim_{n \to \infty} \frac{1}{n} I(X^n; X^n + \boldsymbol{u}^{(n)}) \tag{43}$$

$$= I(X; X + Z) \tag{44}$$

$$= \frac{1}{2} \log\left(1 + \frac{\sigma^2}{\eta^2}\right) \tag{45}$$

$$= \frac{1}{2} \log\left(1 + \frac{1}{\frac{\sigma^2(\sigma - \sqrt{P})^2}{\frac{1}{4}\left(\sigma^2 + (\sigma - \sqrt{P})^2 - D\right)^2} - 1}\right) \tag{46}$$

$$= \frac{1}{2} \log \frac{\sigma^2(\sigma - \sqrt{P})^2}{\sigma^2(\sigma - \sqrt{P})^2 - \frac{1}{4}\left(\sigma^2 + (\sigma - \sqrt{P})^2 - D\right)^2}, \tag{47}$$

where $\boldsymbol{u}^{(n)}$ is uniform over $\Lambda^{(n)}$, and $Z \sim \mathcal{N}(0, \eta^2)$. Here, (43) holds by Zamir et al. (2014, Thm. 5.2.1), and (44) is due to Zamir and Feder (1996, Thm. 3).

The distortion satisfies

$$\lim_{n\to\infty} \frac{1}{n} \mathbb{E}\left[\left\|X^n - \hat{X}^n\right\|^2\right] = \lim_{n\to\infty} \frac{1}{n} \mathbb{E}\left[\left\|X^n - \frac{\sigma - \sqrt{P}}{\sqrt{\sigma^2 + \eta^2}}(X^n + \boldsymbol{u}^{(n)})\right\|_2^2\right] \tag{48}$$

$$= \lim_{n\to\infty} \frac{1}{n} \mathbb{E}\left[\left\|X^n - \frac{\sigma - \sqrt{P}}{\sqrt{\sigma^2 + \eta^2}}(X^n + Z^n)\right\|_2^2\right], \tag{49}$$

where $Z^n \sim \mathcal{N}(0, \eta^2 I)$. Here, (48) holds by crypto lemma, and (49) holds since $X^n$ and $\boldsymbol{u}^{(n)}$ are independent, so the squared norm becomes a sum of second moments of $X^n$ and $\boldsymbol{u}^{(n)}$, and $\mathbb{E}[\|\boldsymbol{u}^{(n)}\|^2] = \mathbb{E}[\|Z^n\|^2]$. Since $X^n$ and $Z^n$ are now i.i.d.,

$$\lim_{n\to\infty} \frac{1}{n} \mathbb{E}\left[\left\|X^n - \hat{X}^n\right\|^2\right] = \mathbb{E}\left[\left(X - \frac{\sigma - \sqrt{P}}{\sqrt{\sigma^2 + \eta^2}}(X + Z)\right)^2\right] \tag{50}$$

$$= \left(1 - \frac{\sigma - \sqrt{P}}{\sqrt{\sigma^2 + \eta^2}}\right)^2 \sigma^2 + \left(\frac{(\sigma - \sqrt{P})^2}{\sigma^2 + \eta^2}\right)\eta^2 \tag{51}$$

$$= \sigma^2 + (\sigma - \sqrt{P})^2 - 2\sigma^2 \frac{\sigma - \sqrt{P}}{\sqrt{\sigma^2 + \eta^2}} \tag{52}$$

$$= \sigma^2 + (\sigma - \sqrt{P})^2 - (\sigma^2 + (\sigma - \sqrt{P})^2 - D) \tag{53}$$

$$= D. \tag{54}$$

For the case when $\sqrt{P} \geq \sigma - \sqrt{|\sigma^2 - D|}$, we use a sequence of DLTCs with $g_a^{(n)}(\boldsymbol{v}) = \boldsymbol{v}$, $g_s^{(n)}(\boldsymbol{v}) = \left(\frac{\sigma^2 - D}{\sigma^2}\right)\boldsymbol{v}$, and a sequence of sphere-bound-achieving lattices $\Lambda^{(n)}$ with second moment $\sigma^2(\Lambda) = \frac{1}{1/D - 1/\sigma^2}$. For the perception constraint, by following the proof of the perception constraint in the previous case, except with $P_{\tilde{Y}^n} = \mathcal{N}(0, (\sigma^2 - D)I_n)$ and $\boldsymbol{z} \sim \mathcal{N}\left(0, \frac{1}{(1/D - 1/\sigma^2)}I_n\right)$, we have that

$$\frac{1}{\sqrt{n}} W_2\left(P_{X^n}, P_{\hat{X}^n}\right) \leq W_2\left(\mathcal{N}(0, \sigma^2), \mathcal{N}(0, \sigma^2 - D)\right) + \epsilon \tag{55}$$

$$= \sigma - \sqrt{|\sigma^2 - D|} + \epsilon \tag{56}$$

$$\leq \sqrt{P} + \epsilon, \tag{57}$$

for any $\epsilon > 0$ and $n$ sufficiently large, where the last step is by the assumption that $\sqrt{P} \geq \sigma - \sqrt{|\sigma^2 - D|}$. The result follows again by continuity of $z \mapsto z^2$. For the rate and distortion constraints of

$$\lim_{n\to\infty} \frac{1}{n} H(Q_{\Lambda^{(n)}}(X^n - \boldsymbol{u}^{(n)})|\boldsymbol{u}^{(n)}) = \max\left\{\frac{1}{2}\log\frac{\sigma^2}{D}, 0\right\}, \tag{58}$$

and

$$\lim_{n\to\infty} \frac{1}{n} \mathbb{E}\left[\left\|X^n - \hat{X}^n\right\|^2\right] = D, \tag{59}$$

the proof follows that of Zamir et al. (2014, Thm. 5.6.1).

$\square$

## D.3 Proof of Theorem 4.5

We now consider the case when the encoder and decoder do not have access to infinite shared randomness. Unlike Thm. 4.3, Thm. 4.5 cannot make use of the additive channel equivalence, and we instead rely on results from lattice Gaussian coding (Ling and Belfiore, 2014). We first introduce several important concepts of lattice Gaussians, then establish several lemmas that will be used to prove Thm. 4.5.

**Definition D.2** (Lattice Gaussian Distribution). A lattice Gaussian random variable $\boldsymbol{y} \sim \mathcal{N}_\Lambda(\boldsymbol{c}, \sigma^2)$ supported on a (shifted) lattice $\Lambda + \boldsymbol{c} \subseteq \mathbb{R}^n$ has PMF

$$q_{\boldsymbol{y}}(\boldsymbol{\lambda}) = \frac{\rho_\sigma(\boldsymbol{\lambda})}{\rho_\sigma(\Lambda + \boldsymbol{c})}, \quad \boldsymbol{\lambda} \in \Lambda + \boldsymbol{c}, \tag{60}$$

where $\rho_\sigma(\boldsymbol{y}) = e^{-\frac{1}{2\sigma^2}\|\boldsymbol{y}\|^2}$ and $\rho_\sigma(\Lambda) = \sum_{\boldsymbol{\lambda} \in \Lambda} \rho_\sigma(\boldsymbol{\lambda})$.

For a more in-depth introduction, see Stephens-Davidowitz (2017). The proof of Thm. 4.5 relies on known results regarding the lattice Gaussian second moment and entropy; see Ling and Belfiore (2014); Regev (2009); Banaszczyk (1993) for details.

Two tools which will be used throughout the proof are the (i) flatness factor of a lattice, and (ii) a vanishing error probability of maximum a posteriori (MAP) decoding of a lattice Gaussian signal sent over an AWGN channel. For (i), the flatness factor (Ling et al., 2014) of a lattice $\Lambda$ is defined as

$$\epsilon_\Lambda(\gamma) := \max_{\boldsymbol{x} \in \mathcal{V}_0(\Lambda)} |V(\Lambda)\rho_{\gamma,\Lambda}(\boldsymbol{x}) - 1|, \tag{61}$$

where $\rho_{\gamma,\Lambda}(\boldsymbol{x}) = \frac{1}{(2\pi\gamma^2)^{n/2}} \sum_{\boldsymbol{\lambda} \in \Lambda} e^{-\frac{\|\boldsymbol{x}-\boldsymbol{\lambda}\|^2}{2\gamma^2}}$. For (ii), suppose we wish to send $\boldsymbol{y} \sim \mathcal{N}_\Lambda(\boldsymbol{c}, \sigma^2 - \nu)$ over an AWGN channel with noise $\boldsymbol{z} \sim \mathcal{N}(0, \nu I_n)$. Ling and Belfiore (2014) show that given the received signal $\tilde{\boldsymbol{x}} = \boldsymbol{y} + \boldsymbol{z}$, the MAP decoder of $\boldsymbol{y}$ given $\tilde{\boldsymbol{x}}$ is given by $Q_{\Lambda-\boldsymbol{c}}(\alpha\tilde{\boldsymbol{x}})$, where $\alpha = \frac{\sigma^2 - \nu}{\sigma^2}$.

The high-level approach in the proof of Thm. 4.5 is to analyze the rate, distortion, and perception when compressing $\tilde{\boldsymbol{x}} = \boldsymbol{y} + \boldsymbol{z}$, following Liu et al. (2021; 2024). This is because $P_{\tilde{\boldsymbol{x}}}$ approximates $\mathcal{N}(0, \sigma^2 I_n)$, with the approximation error given by the flatness factor $\epsilon_\Lambda\left(\sqrt{\frac{(\sigma^2-\nu)\nu}{\sigma^2}}\right)$ (Ling and Belfiore, 2014; Regev, 2009). With a vanishing MAP error probability, $Q_\Lambda(\alpha\tilde{\boldsymbol{x}}) \approx \boldsymbol{y}$, and the overall system can be statistically analyzed using the properties of $\boldsymbol{y}$, which is a lattice Gaussian. We refer the reader to Ling et al. (2014); Ling and Belfiore (2014); Liu et al. (2021) for further details of lattice Gaussian coding.

The next result describes the scaling of the lattice covering radius $r_{\mathrm{cov}}(\Lambda) := \min\{r : \Lambda + \mathcal{B}(\boldsymbol{0}, r) \text{ is a covering of } \mathbb{R}^n\}$, where $\Lambda + \mathcal{B}(\boldsymbol{0}, r)$ is the set composed of spheres of radius $r$ centered at all lattice vectors of $\Lambda$ (Zamir et al., 2014). This allows one to bound the $\ell^2$ error between a vector and its lattice-quantized version by $O(n^{1/2})$.

**Lemma D.3.** *Let $\Lambda$ be a $n$-dimensional lattice with volume $C_1^{n/2}$. Then its covering radius satisfies*

$$r_{\mathrm{cov}}(\Lambda) \le C_2 \sqrt{\frac{\pi e}{2}} \left(\frac{C_1}{\pi}\right)^{1/2} \left[\left(\frac{n}{2}\right)!\right]^{1/n} = O(n^{1/2}), \tag{62}$$

*for a positive constant $C_2$.*

*Proof.* Using results in Zamir et al. (2014, Ch. 3),

$$r_{\mathrm{cov}}(\Lambda) = \rho_{\mathrm{cov}}(\Lambda) \cdot r_{\mathrm{eff}}(\Lambda) \tag{63}$$

$$\le C \sqrt{\frac{\pi e}{2}} \left[\frac{V(\Lambda)}{V_n}\right]^{1/n} \tag{64}$$

$$= C \sqrt{\frac{\pi e}{2}} \left[\frac{C_1^{n/2}}{\frac{\pi^{n/2}}{(n/2)!}}\right]^{1/n} \tag{65}$$

$$= C \sqrt{\frac{\pi e}{2}} \left(\frac{C_1}{\pi}\right)^{1/2} [(n/2)!]^{1/n} \tag{66}$$

$$= O(n^{1/2}), \tag{67}$$

where $C$ is a constant, $\rho_{\mathrm{cov}}$ is the covering efficiency, $r_{\mathrm{eff}}$ is the effective lattice radius, and $V_n$ is the volume of a $n$-dimensional unit ball, following Zamir et al. (2014, Ch. 3). $\qquad\square$

The next lemma bounds the second moment of $Q(\alpha\tilde{\boldsymbol{x}})$, which is used to bound the error terms.

**Lemma D.4.** *Let $\tilde{\boldsymbol{x}} = \boldsymbol{y} + \boldsymbol{z}$, $\boldsymbol{y} \sim \mathcal{N}_\Lambda(\boldsymbol{c}, \sigma^2 - \nu)$ and $\boldsymbol{z} \sim \mathcal{N}(0, \nu I_n)$. Let $\alpha = \frac{\sigma^2 - \nu}{\sigma^2}$. If $\{\Lambda^{(n)}\}_{n=1}^\infty$ is a sequence of lattices that is AWGN-good with vanishing error probability of MAP decoding of $\boldsymbol{y}$ given $\tilde{\boldsymbol{x}}$, then for any $\epsilon > 0$,*

$$\mathbb{E}[\|Q(\alpha\tilde{\boldsymbol{x}})\|^2] \leq \mathbb{E}[\|\boldsymbol{y}\|^2] + \epsilon, \tag{68}$$

*for sufficiently large $n$.*

*Proof.* The second moment of $Q(\alpha\tilde{\boldsymbol{x}})$ satisfies

$$\mathbb{E}\big[\|Q(\alpha\tilde{\boldsymbol{x}})\|^2\big] = \mathbb{E}\Big[\|Q(\alpha\tilde{\boldsymbol{x}})\|^2 \mathbb{1}\{\mathcal{E}^{\complement}\}\Big] + \mathbb{E}\big[\|Q(\alpha\tilde{\boldsymbol{x}})\|^2 \mathbb{1}\{\mathcal{E}\}\big] \tag{69}$$

$$= \mathbb{E}\Big[\|\boldsymbol{y}\|^2 \mathbb{1}\{\mathcal{E}^{\complement}\}\Big] + \mathbb{E}\big[\|Q(\alpha\tilde{\boldsymbol{x}})\|^2 \mathbb{1}\{\mathcal{E}\}\big] \tag{70}$$

$$\leq \mathbb{E}\Big[\|\boldsymbol{y}\|^2\Big] \|\mathbb{1}\{\mathcal{E}^{\complement}\}\|_\infty + \mathbb{E}\big[\|Q(\alpha\tilde{\boldsymbol{x}})\|^2 \mathbb{1}\{\mathcal{E}\}\big] \tag{71}$$

$$= \mathbb{E}\big[\|\boldsymbol{y}\|^2\big] + \mathbb{E}\big[\|Q(\alpha\tilde{\boldsymbol{x}})\|^2 \mathbb{1}\{\mathcal{E}\}\big] \tag{72}$$

$$\leq \mathbb{E}\big[\|\boldsymbol{y}\|^2\big] + \sqrt{\mathbb{E}[\|Q(\alpha\tilde{\boldsymbol{x}})\|^4] \, \mathbb{P}(\mathcal{E})}, \tag{73}$$

where in (71) we use Hölder's inequality, and the last step is by Cauchy-Schwarz. The second term involving the error event vanishes as follows. We have that

$$\|Q(\alpha\tilde{\boldsymbol{x}})\|^4 \leq 8\|Q(\alpha\tilde{\boldsymbol{x}}) - \alpha\tilde{\boldsymbol{x}}\|^4 + 8\|\alpha\tilde{\boldsymbol{x}}\|^4 \tag{74}$$

$$\leq 8\|\boldsymbol{y} - \alpha\tilde{\boldsymbol{x}}\|^4 + 8\|\alpha\tilde{\boldsymbol{x}}\|^4 \tag{75}$$

$$= 8\|(1-\alpha)\boldsymbol{y} - \alpha\boldsymbol{z}\|^4 + 8\|\alpha\boldsymbol{y} + \alpha\boldsymbol{z}\|^4, \tag{76}$$

where (74) is by triangle inequality and Cauchy-Schwarz, and (75) holds since $Q$ finds the closest vector to $\alpha\tilde{\boldsymbol{x}}$. We have that the expectations satisfy

$$\mathbb{E}\big[\|(1-\alpha)\boldsymbol{y} - \alpha\boldsymbol{z}\|^4\big] \tag{77}$$

$$\leq (1-\alpha)^4 \mathbb{E}[\|\boldsymbol{y}\|^4] + \alpha^4 \mathbb{E}[\|\boldsymbol{z}\|^4] + \alpha^2(1-\alpha)^2 \mathbb{E}[\langle \boldsymbol{y}, \boldsymbol{z}\rangle^2] + 2(1-\alpha)^2\alpha^2 \mathbb{E}[\|\boldsymbol{y}\|^2\|\boldsymbol{z}\|^2] \tag{78}$$

$$\leq 3(1-\alpha)^4(\sigma^2 - \nu)^2 n + \alpha^4 \nu^2 n(n+2) + 3\alpha^2(1-\alpha)^2 \mathbb{E}[\|\boldsymbol{y}\|^2] \mathbb{E}[\|\boldsymbol{z}\|^2] \tag{79}$$

$$\leq 3(1-\alpha)^4(\sigma^2 - \nu)^2 n + \alpha^4 \nu^2 n(n+2) + 3\alpha^2(1-\alpha)^2(\sigma^2 - \nu)\nu n^2 \tag{80}$$

$$= O(n^2), \tag{81}$$

and

$$\mathbb{E}\big[\|\alpha\boldsymbol{y} + \alpha\boldsymbol{z}\|^4\big] = \alpha^4 \mathbb{E}\big[\|\boldsymbol{y}\|^4\big] + \alpha^4 \mathbb{E}\big[\|\boldsymbol{z}\|^4\big] + \alpha^4 \mathbb{E}[\langle \boldsymbol{y}, \boldsymbol{z}\rangle^2] + 2\alpha^4 \mathbb{E}\big[\|\boldsymbol{y}\|^2\|\boldsymbol{z}\|^2\big] \tag{82}$$

$$\leq 3\alpha^4(\sigma^2 - \nu)^2 n + \alpha^4 \nu^2 n(n+2) + 3\alpha^4 \mathbb{E}\big[\|\boldsymbol{y}\|^2\|\boldsymbol{z}\|^2\big] \tag{83}$$

$$= 3\alpha^4(\sigma^2 - \nu)^2 n + \alpha^4 \nu^2 n(n+2) + 3\alpha^4(\sigma^2 - \nu)\nu n^2 \tag{84}$$

$$= O(n^2), \tag{85}$$

for $n$ sufficiently large. This implies that

$$\mathbb{E}\big[\|Q(\alpha\tilde{\boldsymbol{x}})\|^4\big] \leq 8\,\mathbb{E}\big[\|(1-\alpha)\boldsymbol{y} - \alpha\boldsymbol{z}\|^4\big] + 8\,\mathbb{E}\big[\|\alpha\boldsymbol{y} + \alpha\boldsymbol{z}\|^4\big] = O(n^2), \tag{86}$$

and therefore

$$\frac{1}{n}\sqrt{\mathbb{E}[\|Q(\alpha\tilde{\boldsymbol{x}})\|^4] \, \mathbb{P}(\mathcal{E})} < \epsilon \tag{87}$$

for $n$ sufficiently large since $\lim_{n\to\infty} \mathbb{P}(\mathcal{E}) = 0$. $\qquad\square$

Before proving Thm. 4.5, we first prove a lemma which describes an achievable RDP region of pre/post-scaled lattice quantization with private dithering on the Gaussian source that makes the relationship between the flatness factors and the scaling $s$ explicit. Ensuring that all flatness factors vanish would imply a single-letter characterization of the achievable RDP region for the near-perfect perception regime, which is what Thm. 4.5 provides.

**Lemma D.5.** *Let $X_1, X_2, \dots \overset{\text{i.i.d.}}{\sim} \mathcal{N}(0, \sigma^2)$. Let $\nu \in (0, \sigma^2)$, $\alpha = \frac{\sigma^2 - \nu}{\sigma^2}$, and $s > 0$, $\beta > 0$ be constants satisfying*

$$(\sigma^2 - \nu) + s^2 \frac{(\sigma^2 - \nu)\nu}{\sigma^2} = \frac{\sigma^2}{\beta^2}. \tag{88}$$

Let $\tilde{\boldsymbol{x}} = \boldsymbol{y} + \boldsymbol{z}$, $\boldsymbol{y} \sim \mathcal{N}_\Lambda(\boldsymbol{c}, \sigma^2 - \nu)$ and $\boldsymbol{z} \sim \mathcal{N}(0, \nu I_n)$. If $\{\Lambda^{(n)}\}_{n=1}^\infty$ is a sequence of lattices that is AWGN-good with vanishing error probability of MAP decoding of $\boldsymbol{y}$ given $\tilde{\boldsymbol{x}}$, and is also quantization-good, then for any $\epsilon > 0$,

$$\frac{1}{n}H(Q_{\Lambda^{(n)}}(\alpha X^n)) \leq \frac{1}{1 - 4\epsilon_z}\left(\frac{1}{2}\log\frac{\sigma^2}{\nu} + \epsilon_h + \epsilon\right), \tag{89}$$

$$\frac{1}{n}\mathbb{E}[\|X^n - \hat{X}^n\|^2] \leq \frac{1}{1 - 4\epsilon_z}\left((1 - \beta)^2(\sigma^2 - \nu) + \nu + s^2\beta^2\frac{(\sigma^2 - \nu)\nu}{\sigma^2} + \epsilon\right), \tag{90}$$

$$\frac{1}{\sqrt{n}}W_2(P_{X^n}, P_{\hat{X}^n}) \leq \sqrt{8\sigma^2\epsilon_u} + \beta\sqrt{8\epsilon_z\left(2\frac{\sigma^2 - \nu}{1 - 4\epsilon_z} + 2s^2\frac{(\sigma^2 - \nu)\nu}{\sigma^2}\right)} + \epsilon, \tag{91}$$

for $n$ sufficiently large, where $\hat{X}^n = \beta(Q_{\Lambda^{(n)}}(\alpha X^n) + s\boldsymbol{u})$, $\boldsymbol{u} \sim \text{Unif}(\mathcal{V}(\Lambda^{(n)}))$. In the above, $\epsilon_h = -\frac{\log(1-\epsilon_1)}{n} + \frac{\pi(t^{-4}+1)\epsilon_1}{n(1-\epsilon_1)}$, and

$$\epsilon_1 := \epsilon_{\Lambda^{(n)}}\left(\frac{\sqrt{\sigma^2 - \nu}}{\sqrt{\frac{\pi}{\pi - t}}}\right), \quad \epsilon_z := \epsilon_{\Lambda^{(n)}}\left(\sqrt{\frac{(\sigma^2 - \nu)\nu}{\sigma^2}}\right), \quad \epsilon_u := \epsilon_{\Lambda^{(n)}}\left(\sqrt{\frac{(\sigma^2 - \nu)\nu s^2}{\sigma^2 + s^2\nu}}\right) \tag{92}$$

are flatness factors of lattices $\Lambda^{(n)}$, with $0 < t < 1/e$.

*Proof.* Since the lattice sequence is AWGN-good, choose the fundamental volume to satisfy $V(\Lambda^{(n)}) = \left(2\pi e\frac{(\sigma^2 - \nu)\nu}{\sigma^2}(1 + \epsilon_2)\right)^{\frac{n}{2}}$, where $\epsilon_2 \xrightarrow{n\to\infty} 0$, following that of Ling and Belfiore (2014). In the following, we denote the quantizers $Q_{\Lambda^{(n)}}$ as $Q$ with the dependence on the lattice implicit.

Define $\mathcal{E} := \{Q(\alpha\tilde{\boldsymbol{x}}) \neq \boldsymbol{y}\}$ as the "error" event of MAP decoding of $\boldsymbol{y}$ given $\tilde{\boldsymbol{x}}$ (Ling and Belfiore, 2014, Lemma 11). By assumption, $\lim_{n\to\infty}\mathbb{P}(\mathcal{E}) = 0$. Finally, since the lattices are sphere-bound-achieving, we have that the lattice NSM satisfies $\lim_{n\to\infty}G(\Lambda^{(n)}) \to \frac{1}{2\pi e}$. The vanishing error probability and sphere-bound-achieving properties of the lattices will be used to establish the convergence.

We first address the perception constraint. By triangle inequality,

$$\frac{1}{\sqrt{n}}W_2(P_{X^n}, P_{\hat{X}^n}) \leq \frac{1}{\sqrt{n}}W_2(P_{X^n}, P_{\beta(Q(\alpha\tilde{\boldsymbol{x}})+s\boldsymbol{u})}) + \frac{1}{\sqrt{n}}W_2(P_{\hat{X}^n}, P_{\beta(Q(\alpha\tilde{\boldsymbol{x}})+s\boldsymbol{u})}). \tag{93}$$

We first bound the second term on the right. By Regev (2009, Claim 3.9) and Ling and Belfiore (2014, Lemma 9), $|dP_{\tilde{\boldsymbol{x}}}(\boldsymbol{x}) - dP_{X^n}(\boldsymbol{x})| \leq 4\epsilon_z dP_{X^n}(\boldsymbol{x}), \forall \boldsymbol{x}$. By the change of variable formula, this implies $|dP_{\alpha\tilde{\boldsymbol{x}}}(\boldsymbol{x}) - dP_{\alpha X^n}(\boldsymbol{x})| \leq 4\epsilon_z dP_{\alpha X^n}(\boldsymbol{x})$. Additionally, we have that for any $\boldsymbol{\lambda} \in \Lambda$,

$$\Pr(Q(\alpha\tilde{\boldsymbol{x}}) = \boldsymbol{\lambda}) = \int_{\mathcal{V}_0 + \boldsymbol{\lambda}} dP_{\alpha\tilde{\boldsymbol{x}}} \leq \int_{\mathcal{V}_0 + \boldsymbol{\lambda}}(1 + 4\epsilon_z)dP_{\alpha X^n} = (1 + 4\epsilon_z)\Pr(Q(\alpha X^n) = \boldsymbol{\lambda}), \tag{94}$$

and $\Pr(Q(\alpha\tilde{\boldsymbol{x}}) = \boldsymbol{\lambda}) \geq (1 - 4\epsilon_z)\Pr(Q(\alpha X^n) = \boldsymbol{\lambda})$ follows similarly. Thus

$$|\Pr(Q(\alpha\tilde{\boldsymbol{x}}) = \boldsymbol{\lambda}) - \Pr(Q(\alpha X^n) = \boldsymbol{\lambda})| \leq 4\epsilon_z\Pr(Q(\alpha X^n) = \boldsymbol{\lambda}). \tag{95}$$

Finally, we have

$$dP_{Q(\alpha\tilde{\boldsymbol{x}})+s\boldsymbol{u}}(\boldsymbol{w}) = \sum_{\boldsymbol{\lambda}\in\Lambda}\Pr(Q(\alpha\tilde{\boldsymbol{x}}) = \boldsymbol{\lambda})dP_{s\boldsymbol{u}}(\boldsymbol{w} - \boldsymbol{\lambda}) \tag{96}$$

$$\leq \sum_{\boldsymbol{\lambda}\in\Lambda}(1 + 4\epsilon_z)\Pr(Q(\alpha X^n) = \boldsymbol{\lambda})dP_{s\boldsymbol{u}}(\boldsymbol{w} - \boldsymbol{\lambda}) \tag{97}$$

$$= (1 + 4\epsilon_z)dP_{Q(\alpha X^n)+s\boldsymbol{u}}(\boldsymbol{w}), \tag{98}$$

where the other direction follows similarly. Thus

$$|dP_{Q(\alpha\tilde{\boldsymbol{x}})+s\boldsymbol{u}}(\boldsymbol{w}) - dP_{Q(\alpha X^n)+s\boldsymbol{u}}(\boldsymbol{w})| \leq 4\epsilon_z dP_{Q(\alpha X^n)+s\boldsymbol{u}}(\boldsymbol{w}). \tag{99}$$

for any $\boldsymbol{w}$. This implies

$$\frac{1}{\beta\sqrt{n}}W_2\big(P_{\hat{X}^n},P_{\beta(Q(\alpha\tilde{\boldsymbol{x}})+s\boldsymbol{u})}\big) = \frac{1}{\sqrt{n}}W_2\big(P_{Q(\alpha X^n)+s\boldsymbol{u}},P_{Q(\alpha\tilde{\boldsymbol{x}})+s\boldsymbol{u}}\big) \tag{100}$$

$$\leq \frac{1}{\sqrt{n}}\sqrt{2\int\|\boldsymbol{w}\|^2|dP_{Q(\alpha\tilde{\boldsymbol{x}})+s\boldsymbol{u}}(\boldsymbol{w})-dP_{Q(\alpha X^n)+s\boldsymbol{u}}(\boldsymbol{w})|d\boldsymbol{w}} \tag{101}$$

$$\leq \sqrt{8\epsilon_z\frac{1}{n}\,\mathbb{E}[\|Q(\alpha X^n)+s\boldsymbol{u}\|^2]}, \tag{102}$$

where (101) is by Villani (2016, Thm. 6.15). To bound the second norm, we first get

$$\frac{1}{n}\,\mathbb{E}[\|Q(\alpha X^n)+s\boldsymbol{u}\|^2] \leq 2\frac{1}{n}\,\mathbb{E}[\|Q(\alpha X^n)\|^2]+2\frac{1}{n}\,\mathbb{E}[\|s\boldsymbol{u}\|^2] \tag{103}$$

$$= 2\frac{1}{n}\,\mathbb{E}[\|Q(\alpha X^n)\|^2]+2s^2\frac{(\sigma^2-\nu)\nu}{\sigma^2}. \tag{104}$$

The second moment of $Q(\alpha X^n)$ satisfies

$$\mathbb{E}[\|Q(\alpha X^n)\|^2] = \sum_{\boldsymbol{\lambda}}\|\boldsymbol{\lambda}\|^2\Pr(Q(\alpha X^n)=\boldsymbol{\lambda}) \tag{105}$$

$$= \sum_{\boldsymbol{\lambda}}\|\boldsymbol{\lambda}\|^2(\Pr(Q(\alpha\tilde{\boldsymbol{x}})=\boldsymbol{\lambda})-\Pr(Q(\alpha X^n)=\boldsymbol{\lambda}))+\sum_{\boldsymbol{\lambda}}\|\boldsymbol{\lambda}\|^2\Pr Q(\alpha\tilde{\boldsymbol{x}})=\boldsymbol{\lambda}) \tag{106}$$

$$\leq 4\epsilon_z\,\mathbb{E}[\|Q(\alpha X^n)\|^2]+\mathbb{E}[\|Q(\alpha\tilde{\boldsymbol{x}})\|^2], \tag{107}$$

where the last step is by (95). Thus for $n$ sufficiently large,

$$\frac{1}{n}\,\mathbb{E}[\|Q(\alpha X^n)\|^2] \leq \frac{1}{1-4\epsilon_z}\frac{1}{n}\,\mathbb{E}[\|Q(\alpha\tilde{\boldsymbol{x}})\|^2] \tag{108}$$

$$\leq \frac{1}{1-4\epsilon_z}(\sigma^2-\nu), \tag{109}$$

where the last step is by Lemma D.4 and Banaszczyk (1993). Combining with (102) and (104),

$$\frac{1}{\sqrt{n}}W_2\big(P_{\hat{X}^n},P_{\beta(Q(\alpha\tilde{\boldsymbol{x}})+s\boldsymbol{u})}\big) \leq \beta\sqrt{8\epsilon_z\left(2\frac{\sigma^2-\nu}{1-4\epsilon_z}+2s^2\frac{(\sigma^2-\nu)\nu}{\sigma^2}\right)}. \tag{110}$$

For the first term on the right of (93), we again apply triangle inequality:

$$\frac{1}{\sqrt{n}}W_2(P_{X^n},P_{\beta(Q(\alpha\tilde{\boldsymbol{x}})+s\boldsymbol{u})}) \leq \underbrace{\frac{1}{\sqrt{n}}W_2(P_{X^n},P_{\beta(\boldsymbol{y}+s\boldsymbol{u})})}_{A}+\underbrace{\frac{1}{\sqrt{n}}W_2(P_{\beta(Q(\alpha\tilde{\boldsymbol{x}})+s\boldsymbol{u})},P_{\beta(\boldsymbol{y}+s\boldsymbol{u})})}_{B}. \tag{111}$$

For term $A$, let $\tilde{\boldsymbol{z}}\sim\mathcal{N}\left(0,\frac{(\sigma^2-\nu)\nu}{\sigma^2}I_n\right)$ and $\boldsymbol{r}\sim\mathcal{N}\left(0,\frac{\sigma^2}{\beta^2}I_n\right)$. Then

$$\frac{1}{\sqrt{n}}W_2(P_{\boldsymbol{r}},P_{\boldsymbol{y}+s\tilde{\boldsymbol{z}}}) \leq \frac{1}{\sqrt{n}}\sqrt{2\int\|\boldsymbol{w}\|^2|dP_{\boldsymbol{r}}-dP_{\boldsymbol{y}+s\tilde{\boldsymbol{z}}}|d\boldsymbol{w}} \tag{112}$$

$$\leq \frac{1}{\sqrt{n}}\sqrt{8\epsilon_u\,\mathbb{E}_{\boldsymbol{r}}[\|\boldsymbol{r}\|^2]} \tag{113}$$

$$= \sqrt{8\epsilon_u\frac{\sigma^2}{\beta^2}}, \tag{114}$$

where (112) is by Villani (2016, Thm. 6.15), and (113) holds since $s, \beta$ satisfy (88) and by applying Ling and Belfiore (2014, Lemma 9). Therefore

$$\frac{1}{\beta\sqrt{n}}W_2(P_{X^n}, P_{\beta(\boldsymbol{y}+s\boldsymbol{u})}) = \frac{1}{\sqrt{n}}W_2(P_{\boldsymbol{r}}, P_{\boldsymbol{y}+s\boldsymbol{u}}) \tag{115}$$

$$\leq \frac{1}{\sqrt{n}}W_2(P_{\boldsymbol{r}}, P_{\boldsymbol{y}+s\tilde{\boldsymbol{z}}}) + \frac{1}{\sqrt{n}}W_2(P_{\boldsymbol{y}+s\tilde{\boldsymbol{z}}}, P_{\boldsymbol{y}+s\boldsymbol{u}}) \tag{116}$$

$$\leq \sqrt{8\epsilon_u \frac{\sigma^2}{\beta^2}} + s\frac{1}{\sqrt{n}}W_2(P_{\boldsymbol{u}}, P_{\tilde{\boldsymbol{z}}}) \tag{117}$$

$$\leq \sqrt{8\epsilon_u \frac{\sigma^2}{\beta^2}} + s\frac{1}{\sqrt{n}}\sqrt{2\frac{(\sigma^2-\nu)\nu}{\sigma^2}D_{\mathsf{KL}}(\boldsymbol{u}||\tilde{\boldsymbol{z}})} \tag{118}$$

$$\leq \sqrt{8\epsilon_u \frac{\sigma^2}{\beta^2}} + \frac{\epsilon}{2\beta}, \tag{119}$$

where (117) holds by Ling and Belfiore (2014, Lemma 9) for $n$ sufficiently large, and data processing inequality for $W_2$ (Santambrogio, 2015, Lemma 5.2), (118) holds by Talagrand (1996), and (119) holds because

$$\frac{1}{n}D_{\mathsf{KL}}(\boldsymbol{u}||\tilde{\boldsymbol{z}}) = \frac{1}{n}\mathbb{E}_{\boldsymbol{u}}[-\log p_{\tilde{\boldsymbol{z}}}(\boldsymbol{u})] - \frac{1}{n}H(\boldsymbol{u}) \tag{120}$$

$$= \frac{1}{n}\left[\frac{1}{2\ln 2\frac{(\sigma^2-\nu)\nu}{\sigma^2}}\mathbb{E}_{\boldsymbol{u}}[\|\boldsymbol{u}\|^2] + \frac{n}{2}\log\left(2\pi\frac{(\sigma^2-\nu)\nu}{\sigma^2}\right)\right] - \frac{1}{n}\log V(\Lambda^{(n)}) \tag{121}$$

$$= G(\Lambda^{(n)}) \cdot 2\pi e\frac{1+\epsilon_2}{2\ln 2} + \frac{1}{2}\log\frac{1}{e(1+\epsilon_2)} \tag{122}$$

$$= \frac{1}{2\ln 2}\Big(G(\Lambda^{(n)}) \cdot 2\pi e(1+\epsilon_2) - 1\Big) - \frac{1}{2}\log(1+\epsilon_2) \tag{123}$$

$$\xrightarrow{n\to\infty} 0, \tag{124}$$

where we use the fact that $\frac{1}{n}\mathbb{E}[\|\boldsymbol{u}\|^2] = G(\Lambda^{(n)}) \cdot 2\pi e \cdot \frac{(\sigma^2-\nu)\nu}{\sigma^2}(1+\epsilon_2)$ and the lattice sequence is sphere-bound-achieving. Thus

$$A = \frac{1}{\sqrt{n}}W_2(P_{X^n}, P_{\beta(\boldsymbol{y}+\boldsymbol{u})}) \leq \sqrt{8\sigma^2\epsilon_u} + \frac{\epsilon}{2}. \tag{125}$$

For term $B$, let us first divide by $\beta$ and analyze the squared 2-Wasserstein; this gives us $\frac{1}{n}W_2^2(P_{Q(\alpha\tilde{\boldsymbol{x}})+s\boldsymbol{u}}, P_{\boldsymbol{y}+s\boldsymbol{u}}) \leq \frac{1}{n}W_2^2(P_{Q(\alpha\tilde{\boldsymbol{x}})}, P_{\boldsymbol{y}})$ by Santambrogio (2015, Lemma 5.2). Let $\pi$ be the coupling between $P_{Q(\alpha\tilde{\boldsymbol{x}})}, P_{\boldsymbol{y}}$ induced by the joint $P_{\tilde{\boldsymbol{x}},\boldsymbol{y}}$; i.e., $\hat{\boldsymbol{y}}, \boldsymbol{y} \sim \pi$ means that $\hat{\boldsymbol{y}} = Q(\alpha(\boldsymbol{y}+\boldsymbol{z}))$ with $\boldsymbol{z} \sim \mathcal{N}(0, \nu I_n)$ as defined above. Then,

$$\frac{1}{n}W_2^2(P_{Q(\alpha\tilde{\boldsymbol{x}})}, P_{\boldsymbol{y}}) = \frac{1}{n}\min_{\pi'\in\Pi(P_{Q(\alpha\tilde{\boldsymbol{x}})}, P_{\boldsymbol{y}})}\mathbb{E}_{\hat{\boldsymbol{y}},\boldsymbol{y}\sim\pi'}\big[\|\hat{\boldsymbol{y}} - \boldsymbol{y}\|^2\big] \tag{126}$$

$$\leq \frac{1}{n}\mathbb{E}_{\hat{\boldsymbol{y}},\boldsymbol{y}\sim\pi}\big[\|\hat{\boldsymbol{y}} - \boldsymbol{y}\|^2\big] = \frac{1}{n}\mathbb{E}_{\boldsymbol{y},\boldsymbol{z}}\big[\|\hat{\boldsymbol{y}} - \boldsymbol{y}\|^2\big] \tag{127}$$

$$= \frac{1}{n}\mathbb{E}_{\boldsymbol{y},\boldsymbol{z}}\Big[\|\hat{\boldsymbol{y}} - \boldsymbol{y}\|^2\Big|\mathcal{E}^{\complement}\Big]\mathbb{P}(\mathcal{E}^{\complement}) + \frac{1}{n}\mathbb{E}_{\boldsymbol{y},\boldsymbol{z}}\big[\|\hat{\boldsymbol{y}} - \boldsymbol{y}\|^2\mathbb{1}\{\mathcal{E}\}\big] \tag{128}$$

$$= \frac{1}{n}\mathbb{E}_{\boldsymbol{y},\boldsymbol{z}}\big[\|\hat{\boldsymbol{y}} - \boldsymbol{y}\|^2\mathbb{1}\{\mathcal{E}\}\big] \tag{129}$$

$$\leq \frac{1}{n}\sqrt{\mathbb{E}[\|\hat{\boldsymbol{y}} - \boldsymbol{y}\|^4]\mathbb{P}(\mathcal{E})} \tag{130}$$

where $\mathcal{E} := \{Q(\alpha\tilde{\boldsymbol{x}}) \neq \boldsymbol{y}\} = \{(\alpha-1)\boldsymbol{y} + \alpha\boldsymbol{z} \notin \mathcal{V}_0(\Lambda)\}$ is the "error" event that quantizing $\alpha\tilde{\boldsymbol{x}}$ does not equal the lattice Gaussian $\boldsymbol{y}$, and the last step is by Cauchy-Schwarz. For (129), we have

that $\mathbb{E}_{\boldsymbol{y},\boldsymbol{z}}\left[\|\hat{\boldsymbol{y}} - \boldsymbol{y}\|^2\Big|\mathcal{E}^{\mathsf{C}}\right] = \mathbb{E}_{\boldsymbol{y},\boldsymbol{z}}\left[\|\hat{\boldsymbol{y}} - \boldsymbol{y}\|^2|\hat{\boldsymbol{y}} = \boldsymbol{y}\right] = 0$. Note that

$$\|\hat{\boldsymbol{y}} - \boldsymbol{y}\| = \|\hat{\boldsymbol{y}} - \alpha\tilde{\boldsymbol{x}} + \alpha\tilde{\boldsymbol{x}} - \boldsymbol{y}\| \leq \|\hat{\boldsymbol{y}} - \alpha\tilde{\boldsymbol{x}}\| + \|\boldsymbol{y} - \alpha\tilde{\boldsymbol{x}}\| = \min_{\boldsymbol{y}'\in\Lambda^{(n)}} \|\boldsymbol{y}' - \alpha\tilde{\boldsymbol{x}}\| + \|\boldsymbol{y} - \alpha\tilde{\boldsymbol{x}}\| \tag{131}$$

$$\leq \|\boldsymbol{y} - \alpha\tilde{\boldsymbol{x}}\| + \|\boldsymbol{y} - \alpha\tilde{\boldsymbol{x}}\| = 2 \cdot \|\boldsymbol{y} - \alpha\tilde{\boldsymbol{x}}\|. \tag{132}$$

This implies that

$$\mathbb{E}\left[\|\hat{\boldsymbol{y}} - \boldsymbol{y}\|^4\right] \tag{133}$$

$$\leq 16\,\mathbb{E}\left[\|\boldsymbol{y} - \alpha\tilde{\boldsymbol{x}}\|^4\right] \tag{134}$$

$$= 16\,\mathbb{E}\left[\|(1-\alpha)\boldsymbol{y} - \alpha\boldsymbol{z}\|^4\right] \tag{135}$$

$$= 16(1-\alpha)^4\,\mathbb{E}[\|\boldsymbol{y}\|^4] + 16\alpha^4\,\mathbb{E}[\|\boldsymbol{z}\|^4] + 64P\alpha^2(1-\alpha)^2\,\mathbb{E}[\langle\boldsymbol{y},\boldsymbol{z}\rangle^2] + 32(1-\alpha)^2\alpha^2\,\mathbb{E}[\|\boldsymbol{y}\|^2\|\boldsymbol{z}\|^2] \tag{136}$$

where we use the fact that $y, z$ are independent. Note that $\frac{1}{n}\,\mathbb{E}[\|\boldsymbol{y}\|^4] \leq 3(\sigma^2 - \nu)^2$ for $n$ sufficiently large (Zhao and Qian, 2024; Micciancio and Regev, 2004), $\mathbb{E}[\|\boldsymbol{z}\|^4] = \nu^2 n(n+2)$, and $\mathbb{E}[\langle\boldsymbol{y},\boldsymbol{z}\rangle^2] \leq \mathbb{E}[\|\boldsymbol{y}\|^2\|\boldsymbol{z}\|^2] \leq \mathbb{E}[\|\boldsymbol{y}\|^2]\,\mathbb{E}[\|\boldsymbol{z}\|^2]$ by Cauchy-Schwarz and independence. Additionally, $\mathbb{E}[\|\boldsymbol{y}\|^2] \leq n(\sigma^2 - \nu)$ by Banaszczyk (1993). Therefore

$$\mathbb{E}\left[\|\hat{\boldsymbol{y}} - \boldsymbol{y}\|^4\right] \leq 48(1-\alpha)^4 \cdot (\sigma^2 - \nu)^2 n + 16\alpha^4 \cdot \nu^2 n(n+2) + 96\alpha^2(1-\alpha)^2\,\mathbb{E}[\|\boldsymbol{y}\|^2]\,\mathbb{E}[\|\boldsymbol{z}\|^2] \tag{137}$$

$$\leq 48(1-\alpha)^4 \cdot (\sigma^2 - \nu)^2 n + 16\alpha^4 \cdot \nu^2 n(n+2) + 96\alpha^2(1-\alpha)^2 n^2(\sigma^2 - \nu)\nu \tag{138}$$

$$= O(n^2). \tag{139}$$

By the choice of $\alpha$, $Q(\alpha\tilde{\boldsymbol{x}})$ computes the MAP estimate of $\boldsymbol{y}$ (Ling and Belfiore, 2014, Prop. 3). Therefore,

$$\frac{1}{n}\sqrt{\mathbb{E}_{\boldsymbol{y},\boldsymbol{z}}[\|\hat{\boldsymbol{y}} - \boldsymbol{y}\|^4]\,\mathbb{P}(\mathcal{E})} \leq \frac{\epsilon^2}{4\beta^2}, \tag{140}$$

for $n$ sufficiently large, since the error of the MAP estimate satisfies $\lim_{n\to\infty}\mathbb{P}(\mathcal{E}) = 0$. Hence

$$\frac{1}{\sqrt{n}}W_2(P_{\beta(Q(\alpha\tilde{\boldsymbol{x}})+s\boldsymbol{u})}, P_{\beta(\boldsymbol{y}+s\boldsymbol{u})}) \leq \frac{\beta}{\sqrt{n}}W_2(P_{Q(\alpha\tilde{\boldsymbol{x}})}, P_{\boldsymbol{y}}) \tag{141}$$

$$\overset{(129)}{\leq} \beta\sqrt{\frac{1}{n}\,\mathbb{E}_{\boldsymbol{y},\boldsymbol{z}}[\|\hat{\boldsymbol{y}} - \boldsymbol{y}\|^2\,\mathbb{1}\{\mathcal{E}\}]} \tag{142}$$

$$\overset{(140)}{\leq} \beta\frac{\epsilon}{2\beta} = \frac{\epsilon}{2}. \tag{143}$$

In conclusion, we have shown

$$\frac{1}{\sqrt{n}}W_2(P_{X^n}, P_{\hat{X}^n}) \leq \sqrt{8\sigma^2\epsilon_u} + \beta\sqrt{8\epsilon_z\left(2\frac{\sigma^2 - \nu}{1 - 4\epsilon_z} + 2s^2\frac{(\sigma^2 - \nu)\nu}{\sigma^2}\right)} + \epsilon. \tag{144}$$

Next, we address the distortion term. We have that

$$\frac{1}{n}\,\mathbb{E}_{X^n,\boldsymbol{u}}\left[\|X^n - \beta(Q(\alpha X^n) - s\boldsymbol{u})\|^2\right] = \mathbb{E}_{\boldsymbol{u}}\left[\underbrace{\frac{1}{n}\int \|\boldsymbol{x} - \beta(Q(\alpha\boldsymbol{x}) - s\boldsymbol{u})\|^2(dP_{X^n}(\boldsymbol{x}) - dP_{\tilde{\boldsymbol{x}}}(\boldsymbol{x}))d\boldsymbol{x}}_{S_1}\right]$$

$$+ \underbrace{\frac{1}{n}\,\mathbb{E}_{\tilde{\boldsymbol{x}},\boldsymbol{u}}\left[\|\tilde{\boldsymbol{x}} - \beta(Q(\alpha\tilde{\boldsymbol{x}}) - s\boldsymbol{u})\|^2\right]}_{S_2}. \tag{145}$$

By Regev (2009, Claim 3.9) and Ling and Belfiore (2014, Lemma 9), $|dP_{\tilde{x}}(x) - dP_{X^n}(x)| \le 4\epsilon_z dP_{X^n}(x), \forall x$. The first term $S_1$ can be written as

$$S_1 \le 4\epsilon_z \frac{1}{n} \int \|x - \beta(Q(\alpha x) - su')\|^2 dP_{X^n} = 4\epsilon_z \frac{1}{n} \mathbb{E}_{X^n}\big[\|X^n - \beta(Q(\alpha X^n) - su')\|^2\big], \tag{146}$$

for any $u'$ and therefore

$$\frac{1}{n} \mathbb{E}_{X^n, u}\big[\|X^n - \beta(Q(\alpha X^n) - su)\|^2\big] \le \frac{1}{1 - 4\epsilon_z} S_2. \tag{147}$$

We focus on the $S_2$ term. As before, let $\mathcal{E} := \{Q(\alpha\tilde{x}) \ne y\}$ be the "error" event. Then

$$S_2 = \frac{1}{n} \mathbb{E}\big[\|\tilde{x} - \beta(Q(\alpha\tilde{x}) - su)\|^2 \mathbb{1}\{\mathcal{E}^{\complement}\}\big] + \frac{1}{n} \mathbb{E}\big[\|\tilde{x} - \beta(Q(\alpha\tilde{x}) - su)\|^2 \mathbb{1}\{\mathcal{E}\}\big] \tag{148}$$

$$\le \frac{1}{n} \mathbb{E}\big[\|\tilde{x} - \beta(Q(\alpha\tilde{x}) - su)\|^2 \mathbb{1}\{\mathcal{E}^{\complement}\}\big] + \frac{1}{n} \sqrt{\mathbb{E}[\|\tilde{x} - \beta(Q(\alpha\tilde{x}) - su)\|^4]\,\mathbb{P}(\mathcal{E})}, \tag{149}$$

where we use Cauchy-Schwarz. Note that the term with the 4-th moment satisfies

$$\mathbb{E}\big[\|\tilde{x} - \beta(Q(\alpha\tilde{x}) - su)\|^4\big] \tag{150}$$

$$\le 8(1 - \beta\alpha)\,\mathbb{E}\big[\|\tilde{x}\|^4\big] + 8\beta\,\mathbb{E}\big[\|Q(\alpha\tilde{x}) - \alpha\tilde{x}\|^4\big] + 8\beta\,\mathbb{E}\big[\|su\|^4\big]. \tag{151}$$

The second term is $O(n^2)$ by following (136), and the third term satisfies

$$8\beta\,\mathbb{E}\big[\|su\|^4\big] \le 8\beta s^4 r_{\text{cov}}^4(\Lambda^{(n)}) = O(n^2) \tag{152}$$

by Lemma D.3. For the first term,

$$\mathbb{E}[\|\tilde{x}\|^4] = \mathbb{E}\big[\|y + z\|^4\big] \tag{153}$$

$$\le \mathbb{E}[\|y\|^4] + \mathbb{E}[\|z\|^4] + 6\,\mathbb{E}[\|y\|^2]\,\mathbb{E}[\|z\|^2] \tag{154}$$

$$\le 3(\sigma^2 - \nu)^2 n + \nu^2 n(n + 2) + 6(\sigma^2 - \nu)\nu n^2 \tag{155}$$

$$= O(n^2) \tag{156}$$

for $n$ sufficiently large, by the independence of $y$ and $z$, and using 4th moment results on lattice Gaussians from Zhao and Qian (2024); Micciancio and Regev (2004). Since $\lim_{n\to\infty} \mathbb{P}(\mathcal{E}) = 0$, we have that $\frac{1}{n}\sqrt{\mathbb{E}[\|\tilde{x} - \beta(Q(\alpha\tilde{x}) - su)\|^4]\,\mathbb{P}(\mathcal{E})} < \frac{\epsilon}{2}$ for $n$ sufficiently large. Thus

$$S_2 \le \frac{1}{n} \mathbb{E}\big[\|\tilde{x} - \beta(Q(\alpha\tilde{x}) - su)\|^2 \mathbb{1}\{\mathcal{E}^{\complement}\}\big] + \frac{\epsilon}{2} \tag{157}$$

$$= \mathbb{E}\bigg[\frac{1}{n}\|\tilde{x} - \beta(y - su)\|^2 \mathbb{1}\{\mathcal{E}^{\complement}\}\bigg] + \frac{\epsilon}{2} \tag{158}$$

$$\le \mathbb{E}\bigg[\frac{1}{n}\|\tilde{x} - \beta(y - su)\|^2\bigg]\|\mathbb{1}\{\mathcal{E}^{\complement}\}\|_\infty + \frac{\epsilon}{2} \tag{159}$$

$$= \frac{1}{n} \mathbb{E}\big[\|\tilde{x} - \beta(y - su)\|^2\big] + \frac{\epsilon}{2} \tag{160}$$

$$= \frac{1}{n} \mathbb{E}\big[\|(1 - \beta)y + z - \beta su\|^2\big] + \frac{\epsilon}{2} \tag{161}$$

$$= (1 - \beta)^2 \frac{1}{n} \mathbb{E}[\|y\|^2] + \frac{1}{n} \mathbb{E}[\|z\|^2] + s^2\beta^2 \frac{1}{n} \mathbb{E}[\|u\|^2] + \frac{\epsilon}{2} \tag{162}$$

$$\le (1 - \beta)^2(\sigma^2 - \nu) + \nu + s^2\beta^2 \cdot G(\Lambda^{(n)}) \cdot 2\pi e \cdot \frac{(\sigma^2 - \nu)\nu}{\sigma^2}(1 + \epsilon_2) + \frac{\epsilon}{2} \tag{163}$$

$$= (1 - \beta)^2(\sigma^2 - \nu) + \nu + s^2\beta^2 \frac{(\sigma^2 - \nu)\nu}{\sigma^2} + \epsilon, \tag{164}$$

for $n$ sufficiently large, where (159) is by Hölder's inequality, and (163) holds by Banaszczyk (1993), and since the lattice second moment satisfies $\frac{1}{n}\mathbb{E}[\|u\|^2] = G(\Lambda^{(n)}) \cdot V(\Lambda^{(n)})^{2/n}$, and the lattice sequence is quantization-good (i.e., sphere-bound-achieving). The result follows by combining (147) and (164).

Finally, we address the rate term. We have

$$\frac{1}{n} H(Q(\alpha X^n)) = \frac{1}{n} \mathbb{E}_{X^n} \left[ -\log dP_{Q(\alpha X^n)} \right] \tag{165}$$

$$= \underbrace{\frac{1}{n} \int -\log\big(dP_{Q(\alpha \boldsymbol{x})}\big)(dP_{X^n}(\boldsymbol{x}) - dP_{\tilde{\boldsymbol{x}}}(\boldsymbol{x}))d\boldsymbol{x}}_{R_1} + \underbrace{\frac{1}{n} \mathbb{E}_{\tilde{\boldsymbol{x}}} \left[ -\log dP_{Q(\alpha \tilde{\boldsymbol{x}})} \right]}_{R_2}. \tag{166}$$

The first term $R_1$ will vanish as $n \to \infty$, due to the following. By Regev (2009, Claim 3.9) and Ling and Belfiore (2014, Lemma 9), we have that $|dP_{\tilde{\boldsymbol{x}}}(\boldsymbol{x}) - dP_{X^n}(\boldsymbol{x})| \le 4\epsilon_z dP_{X^n}(\boldsymbol{x}), \forall \boldsymbol{x}$, by the choice of the lattice sequence $\Lambda^{(n)}$, for $n$ sufficiently large. Therefore,

$$R_1 \le \frac{1}{n} \int -\log dP_{Q(\alpha \boldsymbol{x})} |dP_{X^n}(\boldsymbol{x}) - dP_{\tilde{\boldsymbol{x}}}(\boldsymbol{x})| d\boldsymbol{x} \tag{167}$$

$$\le 4\epsilon_z \frac{1}{n} \int -\log(dP_{Q(\alpha \boldsymbol{x})}) dP_{X^n}(\boldsymbol{x}) d\boldsymbol{x} \tag{168}$$

$$= 4\epsilon_z \frac{1}{n} H(Q(\alpha X^n)), \tag{169}$$

which implies that

$$\frac{1}{n} H(Q(\alpha X^n)) \le \frac{1}{1 - 4\epsilon_z} R_2, \tag{170}$$

for $n$ sufficiently large.

For $R_2$, this is the per-dimension entropy of $Q(\alpha \tilde{\boldsymbol{x}})$. Let $p(\boldsymbol{\lambda}) := \Pr(Q(\alpha \tilde{\boldsymbol{x}}) = \boldsymbol{\lambda})$ be the PMF of $Q(\alpha \tilde{\boldsymbol{x}})$ supported on $\boldsymbol{\lambda} \in \Lambda$, and let $q_{\boldsymbol{y}}(\boldsymbol{\lambda})$ be the lattice Gaussian PMF of $\boldsymbol{y} \sim \mathcal{N}_\Lambda(\boldsymbol{0}, \sigma^2 - \nu)$. Then

$$H(Q(\alpha \tilde{\boldsymbol{x}})) = -\sum_{\boldsymbol{\lambda} \in \Lambda} p(\boldsymbol{\lambda}) \log p(\boldsymbol{\lambda}) \tag{171}$$

$$\le -\sum_{\boldsymbol{\lambda} \in \Lambda} p(\boldsymbol{\lambda}) \log q_{\boldsymbol{y}}(\boldsymbol{\lambda}) \tag{172}$$

$$= \sum_{\boldsymbol{\lambda} \in \Lambda} p(\boldsymbol{\lambda}) \left[ \frac{1}{2(\sigma^2 - \nu)} \|\boldsymbol{\lambda}\|^2 + \log \rho_{\sqrt{\sigma^2 - \nu}}(\Lambda) \right] \tag{173}$$

$$= \frac{1}{2(\sigma^2 - \nu)} \mathbb{E}\big[ \|Q(\alpha \tilde{\boldsymbol{x}})\|^2 \big] + \log \rho_{\sqrt{\sigma^2 - \nu}}(\Lambda). \tag{174}$$

Combining, we have that

$$R_2 = \frac{1}{n} H(Q(\alpha \tilde{\boldsymbol{x}})) \tag{175}$$

$$\le \frac{1}{2(\sigma^2 - \nu)} \frac{1}{n} \mathbb{E}\big[ \|Q(\alpha \tilde{\boldsymbol{x}})\|^2 \big] + \frac{1}{n} \log \rho_{\sqrt{\sigma^2 - \nu}}(\Lambda) \tag{176}$$

$$\le \frac{1}{2(\sigma^2 - \nu)} \left[ \frac{1}{n} \mathbb{E} \|\boldsymbol{y}\|^2 + 2\epsilon(\sigma^2 - \nu) \right] + \frac{1}{n} \log \rho_{\sqrt{\sigma^2 - \nu}}(\Lambda) \tag{177}$$

$$= \frac{1}{2(\sigma^2 - \nu)} \frac{1}{n} \mathbb{E}\big[ \|\boldsymbol{y}\|^2 \big] + \frac{1}{n} \log \rho_{\sqrt{\sigma^2 - \nu}}(\Lambda) + \epsilon \tag{178}$$

$$= -\frac{1}{n} \sum_{\boldsymbol{\lambda} \in \Lambda} q_{\boldsymbol{y}}(\boldsymbol{\lambda}) \log q_{\boldsymbol{y}}(\boldsymbol{\lambda}) + \epsilon \tag{179}$$

$$= \frac{1}{n} H(\boldsymbol{y}) + \epsilon, \tag{180}$$

for $n$ sufficiently large, where (177) is by Lemma D.4. Therefore, for $n$ sufficiently large, we have

$$\frac{1}{n}H(Q(\alpha X^n)) = \frac{1}{1-4\epsilon_z}\left[\frac{1}{n}H(\boldsymbol{y}) + \epsilon\right] \tag{181}$$

$$\leq \frac{1}{1-4\epsilon_z}\left[\frac{1}{2}\log\frac{\sigma^2}{\nu(1+\epsilon_2)} + \epsilon_h + \epsilon\right] \tag{182}$$

$$\leq \frac{1}{1-4\epsilon_z}\left[\frac{1}{2}\log\frac{\sigma^2}{\nu} + \epsilon_h + \epsilon\right] \tag{183}$$

where (181) is due to (170) and (180). (182) holds by Ling and Belfiore (2014, Lemma 6), the choice of volume, for $n$ sufficiently large.

$\square$

We now prove Thm. 4.5, which is restated below.

**Theorem D.6** (Optimality of PD-LTC for Gaussian sources (Thm. 4.5 in main text)). *Let* $X_1, X_2, \dots \overset{\text{i.i.d.}}{\sim} \mathcal{N}(0, \sigma^2)$. *For any $D$ satisfying $0 < D \leq 2\sigma^2$, there exists a sequence of PD-LTCs* $\{(g_a^{(n)}, g_s^{(n)}, \Lambda^{(n)})\}_{n=1}^{\infty}$ *such that*

$$\lim_{n\to\infty}\frac{1}{n}H(Q_{\Lambda^{(n)}}(g_a^{(n)}(X^n))) \leq R(D/2, \infty), \tag{184}$$

$$\lim_{n\to\infty}\frac{1}{n}\mathbb{E}[\|X^n - \hat{X}^n\|^2] \leq D, \tag{185}$$

$$\lim_{n\to\infty}\frac{1}{n}W_2^2(P_{X^n}, P_{\hat{X}^n}) = 0, \tag{186}$$

*where* $\hat{X}^n = g_s^{(n)}(Q_{\Lambda^{(n)}}(g_a^{(n)}(X^n)) + s\boldsymbol{u})$, $\boldsymbol{u} \sim \text{Unif}(\mathcal{V}(\Lambda^{(n)}))$, *and* $s = \frac{\sigma}{\sqrt{\sigma^2 - D/2}}$.

*Proof.* Set $s = \frac{\sigma}{\sqrt{\sigma^2 - \nu}}$, $\beta = 1$, and $\nu = \frac{D}{2}$, which satisfies (88). Choose the lattices $\Lambda^{(n)}$ as polar lattices (Liu et al., 2021; 2024). By the choice of $s$, $\epsilon_u = \epsilon_z = \epsilon_{\Lambda^{(n)}}\left(\sqrt{\frac{(\sigma^2-\nu)\nu}{\sigma^2}}\right)$, and therefore both flatness factors will vanish exponentially fast as $n \to \infty$ by Liu et al. (2021, Prop. 1). Additionally, by using the fact that $\epsilon_1 \leq \epsilon_{\Lambda^{(n)}}\left(\sqrt{\frac{(\sigma^2-\nu)\nu}{\sigma^2}}/\sqrt{\frac{\pi}{\pi-t}}\right)$, and since $\epsilon_z$ vanishes, taking $t \to 0$ results in a vanishing $\epsilon_1$ and therefore vanishing $\epsilon_h$; see Ling and Belfiore (2014, Sec. III). By Liu et al. (2019), polar lattices are AWGN-good and have exponentially decaying MAP error probability for the $\tilde{\boldsymbol{x}} = \boldsymbol{y} + \boldsymbol{z}$ AWGN channel with lattice Gaussian input, at any signal-to-noise ratio (and therefore any $\nu \in (0, \sigma^2)$). By Liu et al. (2024), the polar lattices are also quantization-good. Therefore, by Lemma D.5 we have that the rate, distortion, and perception satisfy

$$\lim_{n\to\infty}\frac{1}{n}H(Q_{\Lambda^{(n)}}(g_a^{(n)}(X^n))) \leq \frac{1}{2}\log\frac{2\sigma^2}{D}, \tag{187}$$

$$\lim_{n\to\infty}\frac{1}{n}\mathbb{E}[\|X^n - \hat{X}^n\|^2] \leq D, \tag{188}$$

$$\lim_{n\to\infty}\frac{1}{n}W_2^2(P_{X^n}, P_{\hat{X}^n}) = 0, \tag{189}$$

when the transforms are chosen to be $g_a^{(n)}(\boldsymbol{v}) = \alpha\boldsymbol{v}$ and $g_s^{(n)}(\boldsymbol{v}) = \beta\boldsymbol{v}$. $\square$

**Remark D.7.** In the proof of Lemma D.5 (and therefore Thm. 4.5), the condition in (88) ensures that the distribution of $\hat{X}^n$ is approximately Gaussian with covariance $\sigma^2 I_n$, and the closeness of this approximation is quantified by $\epsilon_u$. This essentially controls how well the perception constraint can be enforced, which is why $s$ shows up in the flatness factor. One can see that $s$ needs to be sufficiently large to ensure $\epsilon_u = \epsilon_z$ which guarantees $\epsilon_u$ vanishes by Liu et al. (2021, Prop. 1). If $s$ is too small, $\epsilon_u > \epsilon_z$ and it is not guaranteed it will vanish. We see that the choice of $s = \frac{\sigma}{\sqrt{\sigma^2 - D/2}} > 1$ indicates a dither $s\boldsymbol{u}$ that "bleeds" outside the lattice cell at low rates is sufficient to ensure a vanishing $\epsilon_u$ and therefore vanishing perception. This seems to imply it is not possible to choose a smaller $s$ than

the one chosen and still make $\epsilon_u$ vanish; if this were possible, it would imply that a near-perfect perception rate-distortion tradeoff below $R(D/2, \infty)$ is achievable, which would contradict results in the RDP literature that $R(D/2, \infty)$ is the lower bound (Yan et al., 2021; Wagner, 2022; Chen et al., 2022).

On the other hand, $\epsilon_z$ is used to approximate $P_{X^n} \approx P_{\tilde{x}}$, and $\epsilon_h$ is used to approximate the entropy rate of $y$; as shown in the proof, a vanishing $\epsilon_z$ implies a vanishing $\epsilon_h$.

**Remark D.8.** The sequence of PD-LTCs in Thm. 4.5 that satisfy the near-perfect perception constraint in (16) can be upgraded to a sequence of codes satisfying a perfect perception constraint of $P_{X^n} = P_{\hat{X}^n}$ with the same asymptotic rate and distortion, by following the "coupling argument" of Saldi et al. (2015). That is, if $\pi_{X^n, \hat{X}^n}$ is the coupling that satisfies $\frac{1}{n}W_2^2(P_{X^n}, P_{\hat{X}^n}) = \epsilon$, then one can use $\bar{X}^n \sim \pi_{X^n|\hat{X}^n}$ as the reconstruction instead of $\hat{X}^n$. This would ensure $P_{\bar{X}^n} = P_{X^n}$ and the distortion $\frac{1}{n}\mathbb{E}[\|X^n - \bar{X}^n\|^2] \leq \frac{1}{n}\mathbb{E}[\|X^n - \hat{X}^n\|^2] + \epsilon$. Since this change only affects the decoder, the rate remains the same as before.

**Remark D.9.** The choice of polar lattices in the proof of Thm. 4.5 may not be necessary, and other lattice families may also work. The essential requirements are that both $\epsilon_u$ and $\epsilon_z$ vanish, AWGN-goodness, quantization-goodness, and vanishing MAP error probability for the $\tilde{x} = y + z$ AWGN channel. While the latter three can be simultaneously satisfied by other lattice families, such as the mod-$p$ lattices (Loeliger, 1997), to the best of the authors' knowledge, the only currently known lattice family with known results on vanishing $\epsilon_u$ and $\epsilon_z$ is the polar lattice. This does not preclude the existence of other lattice families that may satisfy all the aforementioned requirements.

## E   Empirical Evaluation of Sliced Wasserstein

Here, we assess how accurate we can estimate the squared 2-Wasserstein Distance $W_2^2(P, Q)$ with the sliced Wasserstein distance $\mathsf{SW}_2^2(P, Q)$ (Bonneel et al., 2015). Let $P = \mathcal{N}(\mathbf{1}, I_n)$, and $Q = \mathcal{N}(\mathbf{0}, 2I_n)$. Then $\frac{1}{n}W_2^2(P, Q) = 2$. Shown in Table 2, 1, sliced Wasserstein provides fairly accurate estimate of the true Wasserstein for Gaussian samples, where $N$ is the number of samples. This supports the use of sliced Wasserstein as a proxy for the Wasserstein distance in our experiment surrounding the Gaussian source (as we would expect the reconstruction distribution to be near-Gaussian). Therefore, the theoretical bounds are a meaningful comparison, as they align with the operational quantities of the corresponding coding theorem.

|  | $N$ | Estimate | Std. Error |
|---|---|---|---|
| | 100 | 2.129 | 0.282 |
| $n = 8$ | 1000 | 1.999 | 0.187 |
| | 5000 | 2.004 | 0.158 |
| | 10000 | 1.994 | 0.164 |
| | 100 | 2.118 | 0.235 |
| | 1000 | 2.017 | 0.192 |
| $n = 24$ | 5000 | 2.005 | 0.188 |
| | 10000 | 2.008 | 0.184 |

Table 1: Estimating $W_2^2$ using Sliced-Wasserstein with 50 projections.

|  | $N$ | Estimate | Std. Error |
|---|---|---|---|
| | 100 | 2.142 | 0.329 |
| $n = 8$ | 1000 | 1.987 | 0.299 |
| | 5000 | 1.999 | 0.274 |
| | 10000 | 2.009 | 0.276 |

Table 2: Estimating $W_2^2$ using Sliced-Wasserstein with 20 projections.

