# OpenReview forum: "Optimal Neural Compressors for the Rate-Distortion-Perception Tradeoff"
_NeurIPS.cc/2025/Conference — NeurIPS 2025 spotlight_

### Official Review · Reviewer_MW7C · 2025-06-30

**Clarity:** 3
**Significance:** 3
**Originality:** 3
**Rating:** 5
**Confidence:** 4

**Summary:**

This paper introduces a neural compression models that aim to the theoretical optimum under the rate-distortion-perception (RDP) tradeoff. Building on recent non-constructive RDP coding theorems, the authors propose constructive implementations leveraging lattice-based transform coding with varying degrees of shared randomness. Their framework includes deterministic LTC, private and shared dither variants (PD-LTC and SD-LTC), and quantized shared dither (QSD-LTC), which are theoretically motivated and experimentally evaluated. Via experiments, this work demonstrates how shared randomness on lattice quantization affect RDP performance.

**Questions:**

1)	In the QSD setting, was the shared randomness rate (Rc = 1.0 or 1.58) included in the total rate?

2)	I understand that the proposed method uses a much more structured form of randomness compared to approaches based on pseudo-random seeds. In addition, the authors claim that the seed-sharing approach using k bits is not suitable for high-dimensional data due to the limitation of having only 2^k distinct seeds. If multiple groups of data dimensions share the same pseudo-random seed, would the proposed method still show a clear advantage? For example, in the case of 2^(k+2)-dimensional data, it seems possible to divide the data into 2^2 groups and assign a shared pseudo-random seed to each group based on a predefined rule, thereby constructing group-wise shared dithers within the lattice space. I am curious whether the proposed method demonstrates a clear benefit compared to such a simple implementation.

**Ethical Concerns:**

["NO or VERY MINOR ethics concerns only"]

**Final Justification:**

Thank you for your thorough response. Many of the reviewer’s concerns have been addressed through the rebuttal, and therefore, I am updating my rating to 5 accordingly.

**Limitations:**

Yes.

**Paper Formatting Concerns:**

None.

**Quality:**

3

**Strengths And Weaknesses:**

[Strength]

Overall, the paper is well-structured and contributes both theoretically grounded and empirically validated insights into the design of RDP-efficient compressors. Particularly, this work effectively bridges the gap between theory and practice by demonstrating how lattice packing and shared randomness affect RDP performance on both synthetic and real-world datasets.

[Weakness]

1)	However, one notable limitation is that, although the authors claim the main advantage of their method over existing approaches, such as RCC, lies in its practicality, the experimental validation is conducted only on the limited, low-dimensional real-world datasets. While the authors mention this as future work, the paper would have been significantly stronger if such validation had been included.

2)	While the proposed architecture is theoretically motivated by the RDP optimality framework, the empirical results suggest that even with infinite shared randomness (i.e., SD-LTC), there remains a non-negligible gap from the theoretical lower bound R(D, 0), indicating practical limitations in achieving RDP-optimality. In particular, as shown in Fig. 4, even under high-distortion constraints, the rate achieved by the proposed method appears to exhibit a form of lower-bound behavior. This observation could be a critical limitation, especially considering that recent perceptual codec techniques are primarily focused on improving subjective quality in the low-rate regime.

3)	For synthetic sources, it is reasonable to compare distortion performance under the constraint P = 0. However, for real-world sources, a trade-off between perception and distortion is observed in some data types (particularly speech). A discussion on this phenomenon is needed.

5)	For synthetic sources, the magnitude "s" of the PD scheme is deterministically derived. In contrast, for real-world sources, the authors should clarify how the magnitude "s" was selected.

6)	(minor) In Section 3.2, the input to lattice quantization is y, the output of g_a. Therefore, it would be more appropriate to use the variable y instead of x in the notation. Alternatively, if this reviewer has misunderstood the proposed method, clarification would be appreciated.

---

> ### Author Rebuttal · Authors · 2025-07-30
>
> We thank the reviewer for their helpful feedback and comments, and recognizing the work as bridging the gap between theory and practice. We address the comments in the following.
>
> **Responses to weaknesses:**
>
> W1. Extending the proposed framework to state-of-the-art high-resolution image compressors is an important direction of our current research. The approach described in the current work could be extended to NTC architectures that handle image data. State-of-the-art architectures for RDP would include HiFiC [1] and follow-up works such as ILLM [2], which are architecturally derived from [3]. Therefore, following [4], we can apply lattices product-wise across the channel dimension, without incurring any notable increase in computational complexity due to the use of dithering or lattice quantization. PD-LTC would involve adding a random dither vector at the decoder in the latent space. The advantage it provides would be an ability to enforce tighter perception constraints compared to deterministic codecs, which we showed for the Speech and Physics datasets. We have preliminary results that show an improvement of FID scores with PD-LTC using the integer lattice with the ILLM model for images: at 0.06 bpp, FID reduces from 5.2 to 4.7 while PSNR remains around 29.1, and at 0.12 bpp, FID reduces from 4.1 to to 3.8 while PSNR improves 30.5 to 30.7. The gain is expected to improve for more efficient lattices. Incorporating SD-LTC in state of the art image compression methods such as ILLM requires some adjustments and variations of the vanilla setting discussed in the present paper. This is because state-of-the-art methods such as [2, 3] use hyperprior entropy models where the latent $y$ is first mean-shifted before quantization and entropy-coding: $y$ is quantized as $Q(y-\mu)+\mu$ where $\mu$ is the mean output from the hyperprior model. We believe that understanding the interplay between $\mu$ and the random dither vector is important in analyzing the role of dithering in coordinating the encoder and decoder and investigating the benefit when random dithering is incorporated in such entropy models. As a result, we believe that a full set of results on image sources requires a more careful investigation of how random dithering should be integrated with the hyperprior models, and this deserves its own separate work.
>
> Nonetheless, our current set of experiments on audio and scientific measurement data, which demonstrate performance benefits for moderate-dimensional data, already illustrate the practical benefits of our design for RDP-oriented compression of scientific and audio sources. Moreover, the theoretical results themselves bridge several sub-fields of neural compression and information theory, including dithered lattice quantization, lattice Gaussian coding, rate-distortion-perception theory, and nonlinear/lattice transform coding. We believe the proof techniques and theorems to be of potential independent interest to all these sub-fields, providing additional value to the paper’s contributions.
>
> W2. The theoretical results provided are asymptotic, i.e., to achieve the fundamental limits, a very large dimensional lattice would need to be used. For practical uses, a finite-dimensional lattice must be chosen. The experimental results show that the predicted behaviors are still exhibited by finite-length lattice codes (such as $E_8$), which is enough to significantly bridge the gap to $R(D, P)$ and $R(D/2, \infty)$. Using higher-dimensional lattices would help bring the performance even closer to the fundamental limits. Regarding the low-rate regime in Fig. 4, as mentioned in lines 335-336, this is an artifact due to the limitations regarding the Monte-Carlo integration technique for LTC borrowed from Lei et al (2025); see the ablation studies of Lei et al (2025) for more details. Designing lattice-specific entropy models that can avoid this issue is important future research, and is out of the scope of the current work.
>
> W3. For the real-world sources, we swept $\lambda_1$ (the distortion weight) and kept $\lambda_2$ (the perception weight) fixed to a positive value. This allows us to compare rate-distortion and rate-perception tradeoffs across methods; note that the two plots should be examined jointly together, and that the individual plots do not show the rate-perception (or rate-distortion) performance for a fixed distortion (or perception). We can see that the pairs of figures show how better lattice packing and more shared randomness improve the rate-distortion tradeoff for the perception constraints shown. In addition, (i) deterministic NTC/LTC cannot enforce the perception constraint at low rates compared to PD-NTC/LTC and SD-NTC/LTC, and (ii) SD-LTC achieves a superior rate-distortion tradeoff compared to PD-LTC while also achieving a lower perception. While not directly shown, the perception-distortion tradeoff depends on the rate regimes, and the way the compressors were trained. At large rates, the distortion being small makes a small perception easy to achieve; on Speech (PD-LTC), this would be observed if larger rates were shown. At near-zero rates, the objective is essentially just rate and perception, so the model would prioritize making the perception small. At moderate rates, there is more tension between the perception and distortion constraints, and the tradeoff regime depends on how $\lambda_1$ and $\lambda_2$ relate. If a larger $\lambda_2$ value was chosen for the moderate rate points (e.g., for PD-LTC/NTC), then their perception values would be lowered, at the cost of worse distortion. If $\lambda_1$, $\lambda_2$ were further chosen more particularly, a nicer-looking convex plot would be seen. However, this type of training and figure presentation that we do is consistent with prior RDP works such as [2], and as such the two plots should be examined jointly. We will include this discussion in a future revision of the paper.
>
> W4. For both synthetic and real-world sources, the $s$ parameter is set as a learnable scalar parameter, constrained to be greater than 0.
>
> W5. Regarding notation in Sec. 3.2, we agree it is a bit confusing, and will clarify the notation in a future revision of the paper.
>
> **Response to questions:**
>
> Q1. In QSD-LTC/NTC, $R_c$ is not included in the compression rate $R$. It is simply assumed that a back channel consisting of random bits at a rate of $R_c$ bits per latent dimension is available for use. This matches the setup and assumptions made by the theorems in Section 4, as well as prior work in information theory.
>
>
> Q2. The proposed group-wise dithering is an interesting idea that may have potential in developing alternative efficient dithering methods when finite shared randomness is available. It seems that the proposed group-wise dithering would impose a hierarchical structure on how different dimensions of the dither vector are sampled. Thus, we believe it would implement some sort of nested dithering, which is in line with our proposed QSD-LTC method that is constructed with self-similar nested lattices. However, as mentioned in the last part of Remark B.1 in the paper, QSD-LTC ensures that the finite set of dither vectors chosen are uniformly distributed throughout the coarse lattice cell and thus maximally cover the cell, which is guaranteed by the properties of self-similar nested lattices. It is unclear whether the group-wise approach would select an equally good set of dither vectors. In addition, the example provided for the group-wise approach would still only generate $2^k$ total possible dither vectors of dimension $k+2$ if the "predefined rule" assigning seeds to groups is deterministic. Specifically, in the example above, let $s$ be a realization of the $2^k$ possible seeds (available at encoder/decoder), and $i \in \{0,1,2,3\}$ be the group index. If $f(s, i)$ is the rule that assigns a seed to group $i$, there are still only $2^k$ possible combined assignments $[f(s, 0), f(s, 1), f(s, 2), f(s, 3)]$ that generate dithers for the whole vector of dimension $k+2$, by enumerating $s$. If $f$ was random, then more than $2^k$ possibilities could be generated for the whole vector, but then the decoder would be unable to decode unless it had access to $f$’s randomness, which would necessitate additional shared randomness beyond the $k$ already shared. Thus it seems unlikely the group-wise approach would alleviate the challenges of a fixed $k$-bit random seed in high-dimensional cases. Overall, the group-wise approach could be promising, but it is unclear whether it provides nice properties on the dithers that QSD-LTC provides due to the self-similar nested lattices.
>
>
> **References**
>
> [1] Mentzer, F., Toderici, G. D., Tschannen, M., and Agustsson, E. High-fidelity generative image compression. NeurIPS, 2020.
>
> [2] Muckley, M. J., El-Nouby, A., Ullrich, K., Jégou, H., and Verbeek, J. Improving statistical fidelity for neural image compression with implicit local likelihood models. ICML 2023.
>
> [3] Ballé, J., Minnen, D., Singh, S., Hwang, S. J., and Johnston, N. Variational image compression with a scale hyperprior. ICLR 2018.
>
> [4] E. Lei, H. Hassani, and S. Saeedi Bidokhti. Approaching rate-distortion limits in neural compression with lattice transform coding. 2024.

---

> ### Comment · Reviewer_MW7C · 2025-08-04
> **.**
>
> Thank you for your thorough response. Many of the reviewer’s concerns have been addressed through the rebuttal, and therefore, I am updating my rating to 5 accordingly.

---

### Official Review · Reviewer_jGLB · 2025-07-02

**Clarity:** 3
**Significance:** 3
**Originality:** 3
**Rating:** 5
**Confidence:** 3

**Summary:**

- Authors propose LTC with infinite or no shared randomness, using a shared dither (SD-LTC) or
private dither (PD-LTC) respectively, and describe the benefits of the former.
- The authors propose a discrete dithering scheme, defined via nested lattices, which enables finite randomness to be shared between the encoder and decoder (QSD-LTC). This scheme provides an interpolation between PD- and SD-LTC, thereby enabling control over the rate of shared randomness.
- The authors analyze SD-LTC theoretically and demonstrate its optimality, while studying the performance of PD-LTC, SD-LTC and QSD-LTC on synthetic and real-world sources empirically.

**Questions:**

- Does it have any advantages over other lattice quantization methods? For example, paper [a] discusses image compression using lattice vector quantization. In terms of RD curve efficiency and computational efficiency, what effect will the proposed method have compared to this?

[a] Xi Zhang, et al. “Learning Optimal Lattice Vector Quantizers for End-to-end Neural Image Compression”, NeurIPS 2024

**Ethical Concerns:**

["NO or VERY MINOR ethics concerns only"]

**Final Justification:**

After reading the authors' responses, I have decided to maintain my positive evaluation.

**Limitations:**

yes

**Quality:**

3

**Strengths And Weaknesses:**

Strength
- Fundamental improvements in quantization in neural compression are expected.
- It is interesting to note that LTC using shared randomness improves compression coding efficiency, both theoretically and experimentally (synthetic and real-world sources across multiple domains).
- By treating finite shared randomness (QSD-LTC), the work may bridges the gap between purely theoretical extremes and practical systems.

Weakness
- As this paper covers fundamental content, it is reasonable that the experiments are performed on relatively low-dimensional data. However, in the modern field of deep image compression, datasets such as Kodak and CLIC are often used for evaluation rather than small-scale data such as MNIST.

---

> ### Author Rebuttal · Authors · 2025-07-30
>
> We thank the reviewer for taking the time to provide valuable feedback and comments, and recognizing our work.
>
> Extending the proposed framework to state-of-the-art high-resolution image compressors is an important direction of our current research. The approach described in the current work could be extended to NTC architectures that handle image data. State-of-the-art architectures for RDP would include HiFiC [1] and follow-up works such as ILLM [2], which are architecturally derived from [3]. Therefore, following [4], we can apply lattices product-wise across the channel dimension, without incurring any notable increase in computational complexity due to the use of dithering or lattice quantization. PD-LTC would involve adding a random dither vector at the decoder in the latent space. The advantage it provides would be an ability to enforce tighter perception constraints compared to deterministic codecs, which we showed for the Speech and Physics datasets. We have preliminary results that show an improvement of FID scores with PD-LTC using the integer lattice with the ILLM model for images: at 0.06 bpp, FID reduces from 5.2 to 4.7 while PSNR remains around 29.1, and at 0.12 bpp, FID reduces from 4.1 to to 3.8 while PSNR improves 30.5 to 30.7. The gain is expected to improve for more efficient lattices. Incorporating SD-LTC in state of the art image compression methods such as ILLM requires some adjustments and variations of the vanilla setting discussed in the present paper. This is because state-of-the-art methods such as [2, 3] use hyperprior entropy models where the latent $y$ is first mean-shifted before quantization and entropy-coding: $y$ is quantized as $Q(y-\mu)+\mu$ where $\mu$ is the mean output from the hyperprior model. We believe that understanding the interplay between $\mu$ and the random dither vector is important in analyzing the role of dithering in coordinating the encoder and decoder and investigating the benefit when random dithering is incorporated in such entropy models. As a result, we believe that a full set of results on image sources requires a more careful investigation of how random dithering should be integrated with the hyperprior models, and this deserves its own separate work.
>
> Nonetheless, our current set of experiments on audio and scientific measurement data, which demonstrate performance benefits for moderate-dimensional data, already illustrate the practical benefits of our design for RDP-oriented compression of scientific and audio sources.
>
> Regarding the types of lattice quantization methods, our theoretical results in Section 4 suggest that one needs a sphere-bound-achieving sequence of lattices to be optimal on the Gaussian source. In other words, on the iid Gaussian source, performance should be best when the normalized second moment (NSM) of the lattice is minimized, and therefore other lattice quantizers would be suboptimal (unless they coincide the NSM-optimal quantizer in each dimension). The lattices chosen, such as the moderate dimension ones used in the experiments, or the high-dimensional ones such as the polar lattices (mentioned in Sec 4), all have low complexity CVP and low NSM. For real-world sources, while our experiments in Section 5 suggest that lower NSM lattices provide better overall RDP performance, it is possible that even better RDP performance could be achieved using the methods in paper [a] which optimize the lattice directly while using the Babai rounding estimate to perform approximate CVP, which avoids the exponential lattice vector search, as there are currently no theoretical guarantees on more general sources. Combining the ideas in [a] with our proposed methods could be an interesting area for future research.
>
>
>
> **References**
>
> [1] Mentzer, F., Toderici, G. D., Tschannen, M., and Agustsson, E. High-fidelity generative image compression. NeurIPS, 2020.
>
> [2] Muckley, M. J., El-Nouby, A., Ullrich, K., Jégou, H., and Verbeek, J. Improving statistical fidelity for neural image compression with implicit local likelihood models. ICML 2023.
>
> [3] Ballé, J., Minnen, D., Singh, S., Hwang, S. J., and Johnston, N. Variational image compression with a scale hyperprior. ICLR 2018.
>
> [4] E. Lei, H. Hassani, and S. Saeedi Bidokhti. Approaching rate-distortion limits in neural compression with lattice transform coding. 2024.

---

> > ### Comment · Reviewer_jGLB · 2025-08-06
> >
> > Thank you for your thoughtful response. I will maintain my positive score.

---

### Official Review · Reviewer_GpJK · 2025-07-03

**Clarity:** 4
**Significance:** 3
**Originality:** 3
**Rating:** 5
**Confidence:** 2

**Summary:**

Although the rate–distortion–perception literature is theoretically well developed, it still lacks a practical scheme with rigorous guarantees. This work fills that gap by introducing lattice transform coding (nonlinear transform coding with a lattice-coded bottleneck) evaluated under three shared-randomness regimes: none, infinite, and finite (quantized). The authors prove the scheme is rate–distortion optimal for Gaussian sources and support their theory with experiments on more complex data sets.

**Questions:**

- As noted in lines 15–16 and 81–82, the RDP performance improves with increasing amounts of shared randomness. Isn’t this relationship somewhat expected? Or is there a nontrivial aspect I’m overlooking? If it’s not trivial, it would be helpful to clarify why highlighting this point as part of the contribution is justified.

- Doesn’t the rate–perception curve for MNIST (Figure 5, bottom left) exhibit somewhat unexpected behavior? It appears that increasing the rate does not consistently lead to improved perception, which seems counterintuitive. Could the authors comment on this?

- Could similar optimal theoretical results be established for QSD-LTC of Gaussian sources, analogous to Theorems 4.3 and 4.5? Since QSD is not mentioned in Section 4, it might be helpful for the reader to include a remark discussing why extending such results to QSD-LTC could be challenging—if that is indeed the case.

**Ethical Concerns:**

["NO or VERY MINOR ethics concerns only"]

**Final Justification:**

I have read the authors’ responses and found their explanations convincing on all points, so I have revised my rating accordingly. The paper already presents interesting and relevant results for the community, and the minor concerns I raised have been satisfactorily addressed in the response.

**Limitations:**

yes

**Quality:**

3

**Strengths And Weaknesses:**

Strengths:
- The literature review and problem statement are beautifully written—clear, concise, and comprehensive.
- The theoretical results for Gaussian sources are particularly valuable and of clear interest to the community.

Weakness:
- Unless I’m missing something, the experimental results do not seem to offer significant new insights into the benefits of using LTC over NTC—this has already been demonstrated by Lei et al. (2025). Similarly, the observed improvements with increased shared randomness are expected. Overall, the experimental section doesn’t appear to add much beyond the valuable theoretical contributions presented in Section 4.

- Although line 992 mentions that the code is provided in the supplementary materials, I wasn’t able to find an anonymized link to the code or any supplementary files. I’m not sure if this is an issue with the OpenReview system or if I simply missed it.

---

> ### Author Rebuttal · Authors · 2025-07-30
>
> We thank the reviewer for recognizing the quality of the writing and theoretical contributions of our work.
>
> We would like to first clarify several points of our work related to weakness 1 and question 1. On shared randomness, while it may be expected that more shared randomness should not hurt the performance, it is not immediately clear how shared randomness, infinite or finite, could be used to boost the performance in an efficient/optimal manner as promised by results from information theory. The extent of this performance gain is also unclear in practical settings (finite dimension $n$, real-world sources beyond Gaussian, etc). In the classical rate-distortion (RD) setting, no randomness is needed to achieve optimality. As a result, most neural codecs, whether oriented for RD or rate-distortion-perception (RDP), do not necessarily use shared randomness or randomness at all as a resource. Recent information-theoretic work (see Sec. 2) has shown that not only randomized reconstructions are needed to satisfy the perception constraint, but also _infinite_ shared randomness is needed to achieve the RDP function. Any less shared randomness results in worse performance. However, these works do not provide constructive schemes that one may implement in practice. In contrast, our work provides a practical scheme that provably achieves the optimal tradeoffs for infinite and no shared randomness as derived by the information theory literature. The experimental results show that our proposed construction utilizes the shared randomness in an efficient manner as predicted by prior information theory work and the theory we established. These trends extend to compression of real-world sources in addition to Gaussians. Furthermore, the theory we provide is asymptotic (e.g., in the limit of large dimensions), but the experimental results show that the predicted behaviors are still exhibited by moderate-length lattice codes (such as $E_8$), which is enough to significantly bridge the gap to $R(D, P)$ and $R(D/2, \infty)$. Finally, while a theoretical proof of QSD-LTC (the finite randomness case) is left for future work (see below), the experimental results show that it can nearly achieve the performance of SD-LTC despite not having infinite shared randomness.
>
> Regarding the benefits of LTC over NTC, Lei et al. (2025) consider the RD setting but do not address the RDP setting. As mentioned before, randomized reconstructions are necessary for RDP. It is therefore not immediately clear whether VQ-like coding (which is optimal for RD) is good for RDP; it is further unclear how VQ should be integrated with randomized codecs as VQ is deterministic. Our work reveals that VQ-like coding is indeed good for RDP (Appendix A), and then showed in Section 3 how lattices, which provide VQ-like regions, could be incorporated with shared or private randomness for neural compression via shared/private dithering. Thus, our experimental results confirm that under the RDP setting, SD/PD-LTC is indeed superior to SD/PD-NTC, is able to approach optimality (as predicted by Thms. 4.3, 4.5), and extends to general sources.
>
> Q2. Regarding the MNIST figure, this is not unexpected. The behavior is the result of the RDP tradeoff being a triple tradeoff, the regimes where the perception constraint is active or not, and the way the compressors were trained. To generate the curves for the general sources, we swept $\lambda_1$ (the distortion weight) and kept $\lambda_2$ (the perception weight) fixed to a positive value. At large rates, the distortion being small makes a small perception easy to achieve. At near-zero rates, the objective is essentially just rate and perception, so the model would prioritize making the perception small. At moderate rates, there is more tension between the perception and distortion constraints, and the tradeoff regime depends on how $\lambda_1$ and $\lambda_2$ relate. If a larger $\lambda_2$ value was chosen for the moderate rate points (e.g., for PD-LTC/NTC), then their perception values would be lowered, at the cost of worse distortion. However, the current figure 5 still illustrates our main arguments: (i) deterministic NTC/LTC cannot enforce the perception constraint at low rates compared to PD-NTC/LTC and SD-NTC/LTC, and (ii) SD-LTC achieves a superior rate-distortion tradeoff compared to PD-LTC while also achieving a lower perception. If PD-LTC’s perception were lowered, it would only worsen its rate-distortion performance, but SD-LTC is already outperforming PD-LTC. If $\lambda_1$, $\lambda_2$ were further chosen more particularly, a nicer-looking convex plot would be seen. However, this type of training and figure presentation that we do is consistent with prior RDP works such as [1, 2], and as such the two plots should be examined jointly.
>
> Q3. Regarding QSD-LTC, this is indeed more difficult. Similar to the proof of PD-LTC, QSD-LTC cannot make use of the additive channel equivalence of dithered lattice quantization. Therefore proving a fundamental limit for QSD-LTC would likely require a similar approach to PD-LTC based on lattice Gaussian analysis. However, unlike PD-LTC, it would require further understanding of the behavior of a discrete dither vector that is distributed over the nested lattices. To the best of our knowledge, this type of dithering has not been studied before in the context of lattice Gaussian coding. Therefore the QSD-LTC analysis deserves its own treatment based on new tools developed for dither vectors defined over nested lattices. We will update the paper with a remark reflecting this.
>
> W2. The code was added to an anonymous link, however, it appears that an error occurred during the submission and this was not included in the supplementary material. While it could be added here, the new rebuttal policies disallow sharing of anonymous links, and we apologize for this inconvenience.
>
> **References**
>
> [1] Y. Blau and T. Michaeli. Rethinking lossy compression: The rate-distortion-perception tradeoff. ICML 2019.
>
> [2] Muckley, M. J., El-Nouby, A., Ullrich, K., Jégou, H., and Verbeek, J. Improving statistical fidelity for neural image compression with implicit local likelihood models. ICML 2023.

---

> > ### Comment · Reviewer_GpJK · 2025-08-01
> >
> > Thank you for the clarifying answers. I have revised my score accordingly.

---

> > > ### Author Response · Authors · 2025-08-09
> > >
> > > Thank you for the follow up and for revising the score. We appreciate your feedback and support.

---

### Official Review · Reviewer_RfEa · 2025-07-03

**Clarity:** 3
**Significance:** 3
**Originality:** 3
**Rating:** 5
**Confidence:** 1

**Summary:**

This paper investigates how to effectively use randomness in neural compression to balance the trade-off between rate, distortion, and perception (RDP). The core method is based on Lattice Quantization, introducing randomness via "dithering". The authors clearly present three architectures for scenarios with infinite shared randomness (SD-LTC), no shared randomness (PD-LTC), and a novel scheme for finite shared randomness (QSD-LTC). The paper not only proves the optimality of its methods on an ideal Gaussian source but also experimentally validates that more shared randomness and better lattice structures improve performance. While I am not an expert in vector quantization, I believe this work clearly connects information-theoretic concepts with practical compressor design.

**Questions:**

none

**Ethical Concerns:**

["NO or VERY MINOR ethics concerns only"]

**Final Justification:**

The authors' rebuttal has adequately addressed my concerns, and I am inclined to accept this paper.

**Limitations:**

yes

**Quality:**

4

**Strengths And Weaknesses:**

Strengths:
1.A major strength is the paper's solid theoretical foundation. The authors prove that their proposed methods achieve the theoretical RDP limits under the standard Gaussian source assumption, which provides strong justification for their model design.
2.The paper does an excellent job of explaining the roles of different types of randomness (shared, private, finite) in compression and provides corresponding solutions. The model architectures are intuitive and easy to understand.
3.The experiments effectively support the paper's theory. The results demonstrate that using better lattices (i.e., with better space-packing efficiency) and more shared randomness indeed leads to better RDP performance.

Weaknesses:
The experiments are primarily conducted on relatively small-scale datasets (e.g., MNIST). State-of-the-art image compression techniques are typically applied to high-resolution images with much more complex models. It is therefore unclear how well the proposed methods would perform in these more challenging, real-world scenarios (e.g., Kodak dataset), which limits the assessment of their practical impact.

---

> ### Author Rebuttal · Authors · 2025-07-30
>
> We thank the reviewer for their helpful comments and feedback, and for recognizing the contributions of our work.
>
> Extending the proposed framework to state-of-the-art high-resolution image compressors is an important direction of our current research. The approach described in the current work could be extended to NTC architectures that handle image data. State-of-the-art architectures for RDP would include HiFiC [1] and follow-up works such as ILLM [2], which are architecturally derived from [3]. Therefore, following [4], we can apply lattices product-wise across the channel dimension, without incurring any notable increase in computational complexity due to the use of dithering or lattice quantization. PD-LTC would involve adding a random dither vector at the decoder in the latent space. The advantage it provides would be an ability to enforce tighter perception constraints compared to deterministic codecs, which we showed for the Speech and Physics datasets. We have preliminary results that show an improvement of FID scores with PD-LTC using the integer lattice with the ILLM model for images: at 0.06 bpp, FID reduces from 5.2 to 4.7 while PSNR remains around 29.1, and at 0.12 bpp, FID reduces from 4.1 to to 3.8 while PSNR improves 30.5 to 30.7. The gain is expected to improve for more efficient lattices. Incorporating SD-LTC in state of the art image compression methods such as ILLM requires some adjustments and variations of the vanilla setting discussed in the present paper. This is because state-of-the-art methods such as [2, 3] use hyperprior entropy models where the latent $y$ is first mean-shifted before quantization and entropy-coding: $y$ is quantized as $Q(y-\mu)+\mu$ where $\mu$ is the mean output from the hyperprior model. We believe that understanding the interplay between $\mu$ and the random dither vector is important in analyzing the role of dithering in coordinating the encoder and decoder and investigating the benefit when random dithering is incorporated in such entropy models. As a result, we believe that a full set of results on image sources requires a more careful investigation of how random dithering should be integrated with the hyperprior models, and this deserves its own separate work.
>
> Nonetheless, our current set of experiments on audio and scientific measurement data, which demonstrate performance benefits for moderate-dimensional data, already illustrate the practical benefits of our design for RDP-oriented compression of scientific and audio sources.
>
> **References**
>
> [1] Mentzer, F., Toderici, G. D., Tschannen, M., and Agustsson, E. High-fidelity generative image compression. NeurIPS, 2020.
>
> [2] Muckley, M. J., El-Nouby, A., Ullrich, K., Jégou, H., and Verbeek, J. Improving statistical fidelity for neural image compression with implicit local likelihood models. ICML 2023.
>
> [3] Ballé, J., Minnen, D., Singh, S., Hwang, S. J., and Johnston, N. Variational image compression with a scale hyperprior. ICLR 2018.
>
> [4] E. Lei, H. Hassani, and S. Saeedi Bidokhti. Approaching rate-distortion limits in neural compression with lattice transform coding. 2024.

---

### Decision · Program_Chairs · 2025-09-17

**Decision:**

Accept (spotlight)

**Comment:**

This paper proposed a neural compression models to the theoretical optimum under the rate-distortion-perception tradeoff. It is well-written, well-structured, and provides clear theoretical contributions. All reviewers give clear positive reviews (all clear accept each with a score of 5) and in their final justification there is no clear further issues remained. Then, all four reviewers acknowledge the clear theoretical contribution for the compression, i.e., "solid theoretical foundation" (reviewer RfEa), "theoretical results... particularly valuable and of clear interest" (reviewer GpJK), "fundamental improvements" (reviewer jGLB), "effectively bridges the gap between theory and practice" (reviewer MW7C). Thus, the AC decides to accept this paper.